# MARWA: MULTI-AGENT RETRIEVAL-AUGMENTED FRAMEWORK FOR RELIABLE BIOINFORMATICS WORK-FLOW AUTOMATION

## ABSTRACT

The rapid growth of multi-omics data has driven the expansion of bioinformatics analysis tools. Common bioinformatics tasks often rely on workflows, which link multiple tools into structured pipelines for reproducibility and scalability. Yet, building workflows manually is slow and error-prone, motivating efforts toward automation. However, bioinformatics workflow automation remains difficult due to the need to clarify vague analytical objectives, coordinate heterogeneous tools, and generate intricate tool commands. Despite the potential of large language models (LLMs) to aid bioinformatics workflow recommendation through advanced semantic understanding and logical reasoning, current agent frameworks often rely on one-shot generation, weak tool retrieval solution, and limited evaluation scheme, resulting in fragile workflow automation. We propose **MARWA**, a Multi-Agent Retrieval-augmented framework for reliable bioinformatics Workflow Automation. The framework emphasizes a step-by-step generation process with error handling at each stage to ensure robustness. We introduce a retrieval-augmented framework to strengthen tool command accuracy, which incorporates multi-perspective LLM-augmented descriptions and employs contrastive learning. We further design a two-stage evaluation framework, combining expert-verified execution on 40 curated tasks with large-scale benchmarking on 2,270 tasks using LLM-based evaluation. Our experiments demonstrate that MARWA consistently outperforms baselines in pass rate, workflow quality and scalability. Our work provides a foundation for trustworthy bioinformatics workflow automation. Project Page: https://anonymous.4open.science/r/MARWA-7D30.

## 1 INTRODUCTION

Bioinformatics is an interdisciplinary field that combines computational science, statistics, and biology to analyze large and complex biological datasets through computational and statistical methods (Luscombe et al., 2001; Gauthier et al., 2019; Baxevanis et al., 2020). With the advances in high-throughput biological technologies (Rhoads & Au, 2015), the field is now confronted with a rapid expansion of biological data. This explosive growth has spurred the development of numerous bioinformatics tools, covering diverse fields such as genomics (Lesk, 2017; Bustamante et al., 2011; Lips et al., 2022), structural biology(Orlando et al., 2022; Jones & Thornton, 2022) and evolutionary biology (Sober, 1994; Losos et al., 2013). These tools have further enabled significant advances in personalized medicine (Heinken et al., 2023) and drug discovery (Hemmerling & Piel, 2022).

Due to the diverse requirements for analyzing biological data, such as genome assembly (Sohn & Nam, 2018) and differential expression analysis (Anders & Huber, 2010), bioinformatics tasks cannot be accomplished using a single bioinformatics tool alone. Instead, they depend on multi-step workflows that organize bioinformatics tools in a sequential, flow-based manner(Fig 1).

Traditional workflow construction often relies heavily on manual scripting and command-line operations. With the emergence of new technologies and algorithms, the workflows are getting increasingly complicated (Subramanian et al., 2020; Schlotter et al., 2018).This approach is not only time-consuming and prone to errors but also hinders repeatability. These issues highlight the need for more automated, intelligent, and trustworthy methods to create bioinformatics workflows.

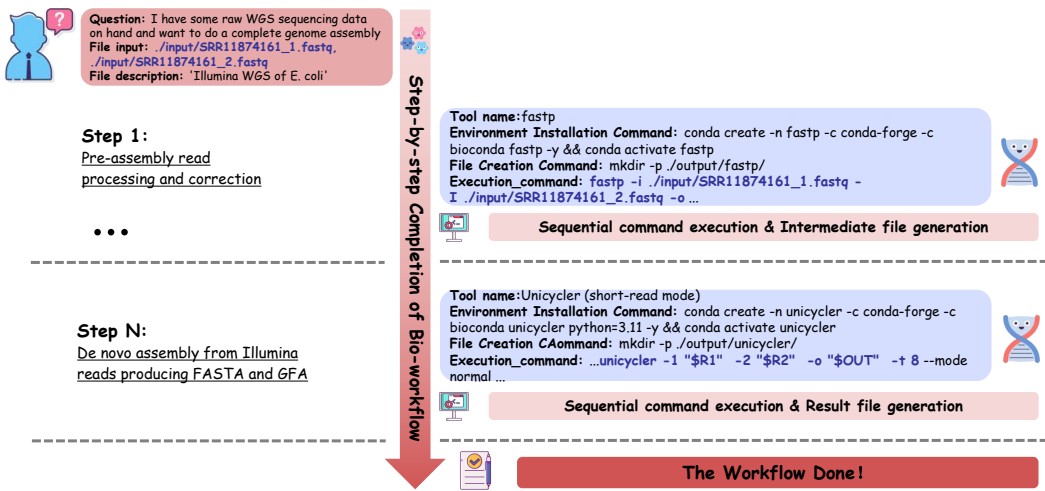

Figure 1: A bioinformatics workflow example for de novo genome assembly. Raw Illumina paired-end reads are processed and quality-controlled (e.g., with fastp) before being assembled into contigs (e.g., with Unicycler), producing final outputs such as FASTA and GFA files. This illustrates the pipeline nature of bioanalysis, where specialized tools are chained together.

Recently, large language models (LLMs) (Zhao et al., 2023; Park et al., 2023; Xi et al., 2025), with their advanced semantic understanding and logical reasoning capabilities, are opening new possibilities for automating bioinformatics workflows. Systems such as AutoBA (Zhou et al., 2023) and BioMaster (Su et al., 2025) show the potential of AI-driven agents, demonstrating their capabilities in automating bioinformatics workflows. However, these approaches remain constrained in three key aspects:

- Adopting one-shot generation strategies makes it struggle to handle vague analytical objectives, coordinate heterogeneous tools, and generate intricate command-line specifications.
- The lack of robust semantic representations for bioinformatics tools makes it difficult to retrieve relevant tools during the retrieval-augmented generation (RAG) (Lewis et al., 2020) process.
- The absence of rigorous evaluation framework results in insufficient validation of the generated workflows' reliability and reproducibility.

To address these challenges, we propose a **M**ulti-**A**gent **R**etrieval-augmented framework for reliable bioinformatics **W**orkflows **A**utomation (MARWA). Our work makes three key contributions:

- We propose MARWA, a step-by-step multi-agent framework that leverages historical context at each stage of workflow construction, thereby enhancing the flexibility and robustness of bioinformatics workflow automation.
- We design a RAG framework that integrates multi-perspective LLM-enhanced tool descriptions with contrastive representation learning, producing discriminative embeddings that significantly improve tool retrieval accuracy and command generation reliability.
- We construct two representative datasets and evaluation standard for bioinformatics workflow automation, comprising a small-scale executable dataset, a large-scale dataset with 2,270 high-quality workflow queries and establish a two-stage evaluation scheme that combines human execution with LLM-based evaluation to ensure rigorous and reproducible benchmarking.

## 2 RELATED WORK

The automation of bioinformatics workflows has undergone a steady evolution, moving from manual construction to intelligent recommendation and, more recently, to LLM-driven automation.

**Workflow Management Systems** Early advances were supported by workflow management platforms such as Galaxy (Jalili et al., 2020)[1], Snakemake [2] and Nextflow (Langer et al., 2025)[3], which provide standardized execution environments and improve reproducibility. Despite these contributions, workflow design still mainly requires manual tool selection and scripting, which renders the process inefficient and prone to mistakes.

**Tool Recommendation** To reduce the burden of tool selection, recommender systems were proposed. For instance, Kumar et al. (2021) employed gated recurrent units (GRU) (Dey & Salem, 2017) neural network in Galaxy to capture higher-order dependencies among tools. Green et al. (2024) further extended this idea by representing workflows as graphs and applying graph neural networks (Wu et al., 2020) with semantic embeddings of tool descriptions. These approaches improve context-aware tool discovery but remain limited to tool-level assistance rather than full workflow automation.

**LLM-Based Workflow Automation** The recent emergence of LLMs has enabled more comprehensive automation (Xi et al., 2025; Zhang et al., 2024; Xiao et al., 2024). AutoBA demonstrated an LLM-based agent can design, implement and execute workflows for diverse omics analyses (Zhou et al., 2023). However, its single-agent design often led to error accumulation in long and complex pipelines. To address this, BioMaster introduced a multi-agent framework with specialized agents for planning, execution, and debugging, combined with RAG of tool knowledge (Su et al., 2025). This multi-agent design improved adaptability and robustness; however, its overall accuracy and reliability were constrained by the limited precision of RAG's embedding matching.

## 3 METHODOLOGY

### 3.1 OVERALL ARCHITECTURE

The overall framework of MARWA is illustrated in Fig 2, and the main algorithm is presented in the Appendix B. It is composed of six cooperative LLM-based expert agents—Analyzing, Planning, Selecting, Generating & Executing, Debugging, and Judging—organized into a closed-loop pipeline. The system is further supported by two auxiliary components: (1) a retrieval module that provides related information about bioinformatics tools, and (2) file system interface access that grounds decisions in the actual execution environment.

The user's input is defined as (1) a list of input files (including their file name and file path), (2) file descriptions (such as format or sequencing type), and (3) the analytical goal (for example, differential expression analysis).

Each agent processes the input and converts it into a structured output, facilitating subsequent parsing. The general agent workflow, as shown in the Fig 3, is outlined below along with a summary of their roles.

**Analyzing** The Analyzing agent refines the user query into a structured task specification. It produces descriptions of the input and output files (including their formats and data types), along with a clarified analytical objective. Appendix A.1 for the prompt of the agent.

**Planning** The Planning agent predicts the next tool to be used in the workflow (see Appendix A.2) based on the refined task and the tools already applied. It provides the tool name, a brief description, and its intended function. It also queries the retrieval module to obtain a set of candidate tools with corresponding descriptions and example commands.

**Selecting** The Selecting agent decides whether to adopt one of the retrieved candidate tools or retain the one proposed by Planning. If a retrieved tool is chosen, its description and command template are adopted; otherwise, the Planning output is used. Appendix A.3 for the prompt of the agent.

---

[1] https://usegalaxy.org/
[2] https://snakemake.github.io/
[3] https://www.nextflow.io/

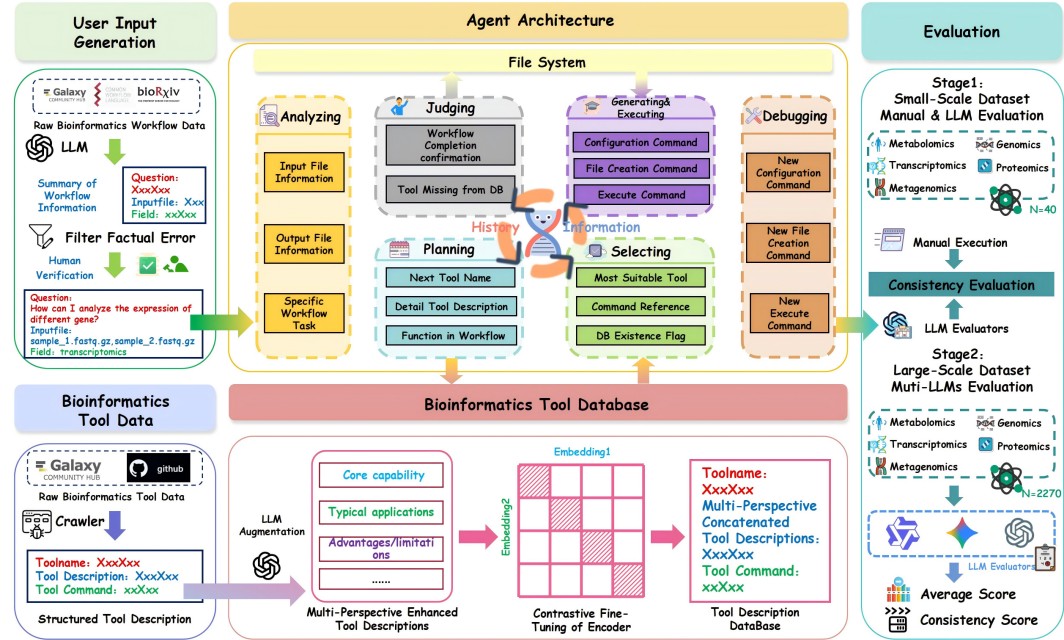

Figure 2: The overall framework of MARWA. The left part shows the generation methods of the data. The middle part illustrates tool retrieval and agent architecture for workflow automation construction. The right part presents workflow evaluation framework.

**Generating & Executing** The Generating & Executing agent constructs executable commands based on the chosen tool and the information available in the system. These commands include environment setup, directory creation, and the actual execution (detail in Appendix A.4). They are dynamically generated and adjusted based on interactions with the live runtime environment, such as detected paths. The commands are then executed, and the success or failure of the execution determines the next step.

**Debugging** If execution fails, the Debugging agent uses the error message to iteratively refine the command set. This process is repeated until the command succeeds or until five attempts have been made. Appendix A.5 for the prompt of the agent.

**Judging** The Judging agent evaluates whether the overall analytical task has been completed (detail in Appendix A.6). The workflow is considered complete only if all required output files are present and every analysis step is fully covered and validated by the tools used. If the task is incomplete, the system loops back to Planning to select the next tool; if complete, the execution terminates. Tools not covered by the retrieval module but successfully executed are recorded, along with their verified commands, to expand the system's tool database.

## 3.2 AUXILIARY COMPONENTS

### 3.2.1 EMBEDDING

Since bioinformatics tool descriptions often come from heterogeneous platforms and vary widely in description length and perspective (Ison et al., 2021), conventional embedding methods struggle to achieve precise semantic alignment. To improve the accuracy and reliability of MARWA in the tool retrieval phase, we design a multi-perspective LLM-augmented strategy and refined through contrastive learning fine-tuning.

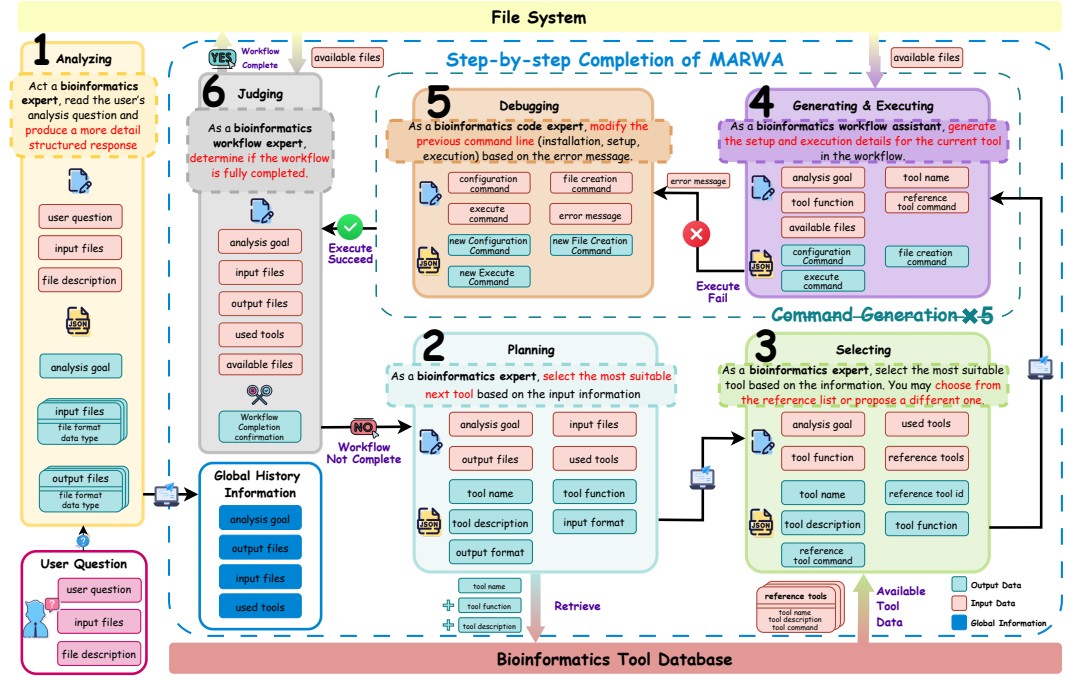

Figure 3: Workflow of MARWA's specialized agents, illustrating the input data, intermediate processing steps, and final output generation.

**Data Sources** We collected raw tool data from open repositories such as Galaxy and GitHub[4]. These unformatted text data were parsed into structured fields including tool name, description, command and parameters. By removing duplicate entries based on names and command hashes, followed by manual verification we have built a curated database containing 3,148 unique tools. This provides a soild foundation for the subsequent stages.

**LLM-based Multi-perspective Enhancement** To enrich the semantic information of the bioinformatics tools, we employ GPT-4 Turbo to generate a diverse range of descriptions for each tool, covering perspectives such as its core capabilities and typical applications (Prompt in Appendix A.7). The resulting augmented descriptions increase semantic diversity and provide complementary views of the same tool, which are stored in structured form for downstream training.

**Contrastive Learning** We adopt BERT (Devlin et al., 2019; Liu et al., 2019) as the encoder, representing each tool description with the [CLS] embedding. We select BERT not only because it remains a widely recognized and reproducible baseline, but also because it provides stable and efficient fine-tuning. This allows us to highlight that the improvements mainly come from our augmentation and contrastive framework rather than from a stronger backbone. Furthermore, its computational efficiency, compared to larger and more complex models, aligns with practical needs for faster iteration and lower resource consumption. Training uses a multi-positive contrastive loss, where augmented descriptions of the same tool serve as positive pairs. For the detailed formulation, see the Appendix D.1.

**Evaluation** To evaluate the effectiveness of the proposed embedding approach, we use a dataset derived from data enhanced by a LLM. The experimental results are shown in Table 1. Our embedding method (LAFT) achieves consistent improvements over all baselines, where BioMaster employs Text-Embedding-3-Large. These results highlight the effectiveness of combining LLM-based augmentation with contrastive learning in capturing the functional semantics of bioinformatics tools. The contribution of this module will be further validated in subsequent ablation studies.

---

[4]https://github.com/

Table 1: Retrieval Performance of baseline embeddings and LLM-augmented fine-tuned model.

| Model | Dim | MRR | Hit@1 | Hit@3 | Hit@10 |
|---|---|---|---|---|---|
| BERT (Devlin et al., 2019) | 768 | 0.1988 | 0.1307 | 0.2010 | 0.2261 |
| all_MiniLM_L6_v2 (Wang et al., 2020) | 384 | 0.4842 | 0.3920 | 0.5477 | 0.5879 |
| Text-Embedding-3-Large | 3072 | 0.5466 | 0.4623 | 0.5729 | 0.6432 |
| PubMedBERT (Gu et al., 2021) | 768 | 0.5798 | 0.5025 | 0.6181 | 0.6533 |
| Qwen3-Embedding-4B | 2560 | 0.6065 | 0.5226 | 0.6382 | 0.6884 |
| bge-en-icl | 4096 | 0.6114 | 0.5141 | 0.6338 | 0.7183 |
| Qwen3-Embedding-8B | 4096 | 0.6458 | 0.5593 | 0.6893 | 0.7458 |
| LAFT | 768 | **0.6686** | **0.5779** | **0.6985** | **0.7638** |

### 3.2.2 FILE SYSTEM INTERFACE

A key challenge in automated workflow generation is bridging the gap between abstract plans generated by LLMs and the real execution environment. To address this, MARWA integrates a file system interface that enables agents to directly query and interact with the underlying directory structure. Specifically, the module provides access to the names and formats of available files, which are then used to guide the construction of subsequent commands.

This interaction yields two main benefits. First, by grounding command generation in the actual file system, the framework reduces errors caused by incorrect file references or incompatible input–output specifications. Second, it ensures that intermediate results are consistently tracked and made available for downstream tools, thereby improving the continuity and robustness of multi-step workflows.

## 4 EXPERIMENTS

### 4.1 DATASETS

To evaluate the effectiveness of our framework, we constructed datasets from multiple real-world bioinformatics workflow repositories, including the Galaxy platform, the Common Workflow Language (CWL)[5] collection, and preprint articles from BioRxiv[6]. These sources were selected because they represent diverse workflow practices and contain detailed applications, covering a wide range of research domains such as metabolomics (Liu & Locasale, 2017), transcriptomics (Lowe et al., 2017), metagenomics (Wooley et al., 2010), genomics (Lesk, 2017) and proteomics (Aslam et al., 2016), thereby ensuring diversity and representativeness.

For further evaluation, we create two datasets of different scales. Table 2 illustrate the statistics of them. The first is a small-scale dataset containing 40 tasks carefully selected by bioinformatics experts (detail in Appendix C.1). Each task is designed for real execution in practical workflows, allowing manual verification and in-depth inspection of model performance. This dataset reflects real-world research demands across major bioinformatics domains, and the task distribution aligned to current practices (Mitchener et al., 2025). We created the second dataset by using GPT-4 Turbo to summarize tasks from raw workflow metadata (data acquisition is provided in Appendix C.2), adopting the roles of a lab researcher, a clinician, and a data engineer. This process produced a large-scale dataset of 2,270 tasks, which has been validated for quality by bioinformatics experts (Appendix A.8 shows the prompt; dataset examples in Appendix C.3). The large-scale dataset maintains a similar domain distribution, further ensuring the representativeness of the small set.

Table 2: Statistics for the datasets.

| | Small | Large |
|---|---|---|
| metabolomics | 4 | 80 |
| transcriptomics | 10 | 600 |
| metagenomics | 8 | 390 |
| genomics | 10 | 900 |
| proteomics | 8 | 300 |
| total | 40 | 2270 |

---

[5]https://view.commonwl.org/workflows/

[6]https://www.biorxiv.org/

## 4.2 EVALUATION FRAMEWORK

To objectively assess the capabilities of different models, we adopted a two-stage evaluation framework. In the first stage, we conducted experiments on the small dataset. Each task was executed manually by domain experts and also simulated by LLMs. We then calculated consistency scores between human execution and model outputs, demonstrating that LLMs can provide reliable evaluations of bioinformatics workflows. The expert-executed results on the small dataset serve as the ground truth, primarily establishing the feasibility of the approach. In the second stage, we scaled up to the large dataset and employed multiple LLMs as evaluators. The large-scale evaluation confirms the trends and demonstrates the method's scalability. These models were used to assign scores to the generated bioinformatics tools and their corresponding command-line, and the final results were reported in terms of both average scores and cross-model consistency. The adoption of LLM-based evaluation is a reasonable yet approximate strategy suitable for large-scale benchmarking. While it confirms the robustness of the method, it does not fully equate to real execution outcomes.

For a comprehensive evaluation, we included both proprietary and open-source models, specifically selecting GPT-4o (GPT-4o) and Gemini-2.5-pro-exp (Gemini2.5) as leading closed-source models, alongside Qwen 2.5 72B-Instruct (Qwen2.5-72B) as a representative open-source alternative. These models were chosen based on prior studies which indicate their strong performance in relevant evaluation tasks (Gu et al., 2024; Liu et al., 2025).

## 4.3 EVALUATION METRICS

We adopted a combination of human-centered and model-based metrics. For manual execution, we used h_Pass@$n$, which measures the success rate of completing a task within $n$ manual execution attempts.

For LLM-based evaluation, we developed a structured scoring template that includes six metrics: (1) **Workflow Completion (Comp)**: Measures whether the workflow achieves the analysis goal (0–3; higher is better). (2) **Workflow Redundancy (Redun)**: Measures whether unnecessary or redundant tools are included(0–3; lower is better). (3) **Installation Accuracy (Inst)**: Correctness of tool installation commands (0–2; higher is better). (4) **Path Accuracy (Path)**: Correctness of file paths used in tool commands (0–2; higher is better). (5) **Parameters Accuracy (Param)**: Correctness of command-line parameters (0–2; higher is better). (6) **Executable Flag**: whether this command be executed successfully (True or False). Score criteria can be found in the Appendix D.3.

The first two metrics operate at the workflow level (prompt in Appendix A.9), while the last four focus on individual tools (prompt in Appendix A.9). These metrics align with common issues in computational method evaluation, making the overall assessment both rigorous and transparent. If a step fails, the system may invoke Debugging to adjust commands and re-evaluate. We define *m_Pass@$n$* as the probability of task success within $n$ LLM-based execution attempts.

To quantify agreement between human and LLM evaluations, we used two measures: (1) **Pass/-Fail Agreement Rate (PFAR)**: The proportion of steps where human execution and LLMs agree on pass/fail outcomes. (2) **Score Agreement Rate (SAR)**: The proportion of instances where the human and LLM scores match exactly for each metric. Formula in Appendix D.2.

We also computed Krippendorff's alpha ($k$) (Krippendorff, 2018; 1970) to assess inter-model agreement among LLM evaluators across all five score metrics, providing a measure of consistency at both workflow and tool levels. In line with established conventions, values above 0.80 indicate reliable agreement, values between 0.67 and 0.80 are considered tentatively acceptable, and values below 0.67 reflect insufficient consistency.

## 4.4 SMALL-SCALE DATASET VALIDATION

We compared MARWA against four baseline methods: LLM-only, AutoBA, ReAct and BioMaster. All models utilized GPT-4 Turbo as the underlying agent to ensure a fair and consistent evaluation. Experimental details can be found in the Appendix D.4.

We evaluated the small dataset using both manual and LLM-based execution with GPT-4o, Gemini2.5, and Qwen2.5-72B. Table 3 reports the average results. More detailed results are provided in

Table 3: The main results of MARWA and different kinds of baselines on the small dataset.

| Method | h_Pass@1 | h_Pass@2 | m_Pass@1 | m_Pass@2 | PFAR | SAR |
|--------|----------|----------|----------|----------|------|-----|
| LLM-only | 0.100 | 0.100 | 0.092 | 0.150 | 0.938 | 0.872 |
| AutoBA | 0.250 | 0.250 | 0.342 | 0.358 | 0.892 | 0.839 |
| ReAct | 0.275 | 0.275 | 0.358 | 0.363 | 0.895 | 0.872 |
| BioMaster | 0.300 | 0.350 | 0.367 | 0.375 | 0.908 | 0.874 |
| MARWA | **0.375** | **0.450** | **0.433** | **0.467** | 0.913 | 0.877 |

Appendix D.5. MARWA surpassed all baseline methods across every evaluation metric, achieving superior performance in both human execution and LLM-based simulation. MARWA's performance advantage can be attributed to its improved capability in selecting appropriate tools, generating more accurate file paths and specifying precise command-line parameters, as clearly demonstrated in the Figure 4. More case studies are provided in Appendix E. A specific running instance of MARWA is provided in the Appendix F. The moderate performance observed across all methods is primarily due to the inherent complexity of real-world bioinformatics workflow automation, which involve multi-step analytical processes, domain-specific tool integration and stringent parameter tuning requirements.

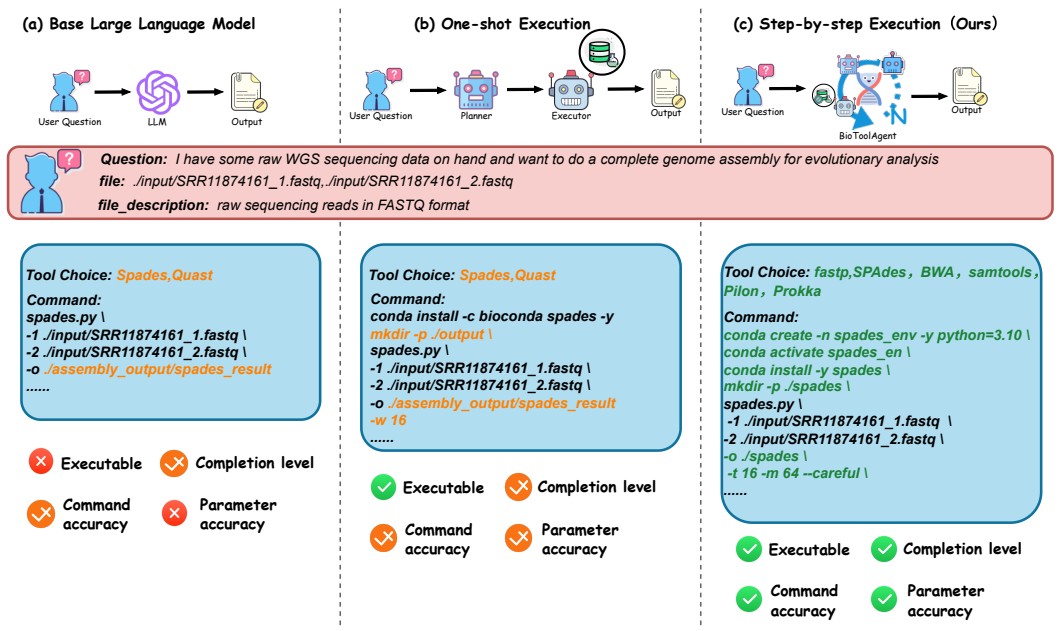

Figure 4: Comparison of bioinformatics workflow automation methods.

## 4.5 LARGE-SCALE DATASET VALIDATION

On the large dataset, MARWA demonstrates consistent superiority across nearly all evaluation metrics. Result in Table 4. A comprehensive analysis of the time consumption is provided in Appendix D.6. We have the following findings.(1) **Workflow Completion and Redundancy**: While MARWA achieves strong workflow completion (Comp: 2.72), the LLM-only approach attains a higher score (2.76) but with more redundancy (Redun: 0.31 vs 0.15). The LLM-only method relies on redundant tools to superficially improve coverage, whereas MARWA emphasizes precision and efficiency through iterative self-correction. Other baselines perform worse in both metrics due to their inability to revise errors in a single pass, leading to accumulated inaccuracies. (2) **Tool Command-Level Reliability**: MARWA achieves the highest path accuracy (Path: 1.78), parameter accuracy (Param: 1.27) and installation correctness (Inst: 1.75). These metrics reflect MARWA's

Table 4: The main results of MARWA and different kinds of baselines on the large dataset.

| Method | Models | Comp | Redun | Inst | Path | Param | m_Pass@1 |
|---|---|---|---|---|---|---|---|
| LLM-only | GPT-4o | 2.83 | 0.29 | 0.61 | 0.19 | 0.76 | 0.11 |
| | Gemini2.5 | 2.75 | 0.31 | 0.44 | 0.12 | 0.72 | 0.11 |
| | Qwen2.5-72B | 2.69 | 0.32 | 0.33 | 0.12 | 0.68 | 0.09 |
| | mean/$k$ | **2.76**/0.77 | 0.31/0.77 | 0.46/0.66 | 0.14/0.77 | 0.72/0.75 | 0.11/0.77 |
| AutoBA | GPT-4o | 2.72 | 0.34 | 1.02 | 0.71 | 0.77 | 0.19 |
| | Gemini2.5 | 2.66 | 0.38 | 0.92 | 0.55 | 0.74 | 0.19 |
| | Qwen2.5-72B | 2.55 | 0.45 | 0.87 | 0.58 | 0.71 | 0.18 |
| | mean/$k$ | 2.64/0.73 | 0.39/0.69 | 0.94/0.76 | 0.61/0.75 | 0.74/0.81 | 0.19/0.73 |
| ReAct | GPT-4o | 2.58 | 0.23 | 1.08 | 1.12 | 1.08 | 0.24 |
| | Gemini2.5 | 2.54 | 0.23 | 1.02 | 1.04 | 1.11 | 0.21 |
| | Qwen2.5-72B | 2.53 | 0.27 | 0.96 | 1.08 | 1.01 | 0.21 |
| | mean/$k$ | 2.55/0.75 | 0.24/0.72 | 1.02/0.71 | 1.08/0.78 | 1.07/0.69 | 0.22/0.74 |
| BioMaster | GPT-4o | 2.62 | 0.22 | 1.74 | 0.63 | 1.15 | 0.25 |
| | Gemini2.5 | 2.58 | 0.24 | 1.63 | 0.52 | 1.08 | 0.24 |
| | Qwen2.5-72B | 2.54 | 0.25 | 1.66 | 0.59 | 1.07 | 0.25 |
| | mean/$k$ | 2.58/0.79 | 0.24/0.73 | 1.68/0.72 | 0.58/0.68 | 1.10/0.70 | 0.25/0.71 |
| MARWA | GPT-4o | 2.74 | 0.15 | 1.77 | 1.79 | 1.28 | 0.41 |
| | Gemini2.5 | 2.71 | 0.14 | 1.74 | 1.77 | 1.26 | 0.40 |
| | Qwen2.5-72B | 2.71 | 0.16 | 1.74 | 1.78 | 1.26 | 0.40 |
| | mean/$k$ | 2.72/0.81 | **0.15**/0.89 | **1.75**/0.76 | **1.78**/0.68 | **1.27**/0.76 | **0.40**/0.76 |

ability to generate reliable tool commands, which is critical for real-world execution. By comparison, all the baselines perform poor on the path accuracy due to the absence of real file system interaction. Although BioMaster incorporates RAG, its embedding mechanism often fails to retrieve relevant and accurate information, resulting in incorrect parameter usage. (3) **Time Efficiency**: MARWA achieves this high accuracy with notable efficiency, as its fast BERT-based retrieval and concise context.

## 4.6 COST-EFFECTIVENESS ANALYSIS

To evaluate the cost of our framework, we analyzed the token consumption for each method, as shown in Table 5. We measured the average input (I-Tokens) and output (O-Tokens) tokens for successful (S) and failed (F) tasks. Based on this data, we calculated the Effective Cost Per Success (ECPS). This metric is derived from the official API pricing of GPT-4 Turbo, our agents' backbone, to reflect the actual monetary expense (see Appendix D.7 for the formula). ECPS represents the average U.S. dollar cost to achieve a single successful workflow, with a lower value indicating superior cost-effectiveness.

Our analysis shows that MARWA achieves the lowest ECPS (0.310), making it the most cost-effective method. MARWA's high success rate prevents costly repeated attempts and debugging cycles, unlike BioMaster (0.693) and ReAct (0.521). This demonstrates that MARWA's design provides a strong balance between high performance and practical efficiency.

## 4.7 ABLATION STUDY

We conducted ablation experiments to evaluate the contribution of each component in MARWA. The results demonstrate that each module plays a distinct role in the process: (1) Removing the retrieval model most severely hurts installation accuracy and overall executability. (2) Disabling the Selecting agent increases workflow redundancy. (3) Removing the Analyzing agent reduces

Table 5: Comparison of token consumption and cost efficiency across different methods.

| Method | I-Tokens (S) | O-Tokens (S) | I-Tokens (F) | O-Tokens (F) | ECPS |
|---|---|---|---|---|---|
| LLM-only | 298 | 689 | 1396 | 2836 | 0.825 |
| AutoBA | 945 | 1043 | 1373 | 2218 | 0.383 |
| ReAct | 4085 | 1329 | 6526 | 1963 | 0.521 |
| BioMaster | 6972 | 1920 | 9203 | 3221 | 0.693 |
| MARWA | 5117 | 1831 | 6529 | 2357 | 0.310 |

Table 6: Ablation study on the small dataset and large dataset.

| Method | Comp | Redun | Inst | Path | Param | m_Pass@1 | h_Pass@1 | h_Pass@2 |
|---|---|---|---|---|---|---|---|---|
| MARWA | 2.72 | 0.15 | 1.75 | 1.78 | 1.27 | 0.40 | 0.375 | 0.45 |
| w/o retrieval | -0.03 | -0.01 | -0.19 | -0.02 | -0.10 | -0.12 | -0.13 | -0.18 |
| w/o Selecting | -0.08 | +0.04 | -0.05 | -0.04 | +0.01 | -0.06 | -0.10 | -0.10 |
| w/o Analing | -0.10 | +0.09 | -0.01 | -0.02 | -0.02 | -0.07 | -0.08 | -0.13 |
| w/o file system | -0.04 | +0.02 | +0.01 | -0.33 | -0.03 | -0.10 | -0.10 | -0.15 |

completion and increases redundancy. (4) Without the file system interface, path accuracy drops sharply. More detailed analyses are provided in the Appendix D.8.

## 5 CONCLUSION

In this paper, we present MARWA, a multi-agent retrieval-augmented framework for reliable bioinformatics workflow automation. MARWA combines a step-by-step generation strategy that decomposes complex tasks into verifiable steps, LLM-augmented retrieval embeddings for precise tool selection and direct file-system interaction to ground commands in the real execution environment. These components reduce error accumulation, improve command and path accuracy and enable reproducible execution. Experiments on diverse real-world datasets show MARWA consistently outperforms strong baselines in execution success and expert-aligned evaluation, offering a practical foundation for trustworthy workflow automation in computational biology.

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

# A   PROMPT TEMPLATES

## A.1   ANALYZING AGENT

---

**Prompt：Analyzing**

You are an assistant for **bioinformatics workflow analysis**.

Your task is to carefully read the user's question about their analysis task,input files and input files' descriptions.
Decompose it into a structured natural language response with the following sections:

1. **Input files**
   - List each input file mentioned by the user.
   - For each file, describe:
     - file name (if provided, otherwise say "not specified")
     - file format (e.g., FASTQ, BAM, VCF, mzML)
     - data type (e.g., raw sequencing reads, aligned reads, variant calls, proteomics spectra)
     - whether it is paired-end (true/false/unknown)
2. **Output files**
   - Describe the expected output files.
   - Include file format (e.g., VCF, TSV, abundance table, PDF report)
       and data type (e.g., variants, gene expression matrix, species abundance).
   - If the user did not specify, infer the most common output for the analysis goal.
3. **Analysis goal**
   - Provide a detailed sentence describing the intended analysis, including:
     - starting point (input files)
     - main processing steps (e.g., quality control, alignment, variant calling)
     - desired outcome (the type of result the user wants)
Rules:
- Always extract actual file names if provided.
- If information is missing, clearly state it as "not specified" or "unknown".
- The output must be well-structured natural language, divided into the three sections above.
- The output must be in **JSON** format as follow:
{"**input_files**": [{
        "file_name": "sample1.fastq | null, // the provided file name, if available
        "file_format": "FASTQ", // e.g., FASTQ, BAM, VCF, mzML
        "data_type": "raw sequencing reads", // e.g., raw sequencing reads, aligned reads
        "paired_end": true | false | null // true/false/null
        }],
"**output_files**": [{
        "file_format": "VCF", // e.g., VCF, TSV, abundance table, PDF report
        "data_type": "variants" // e.g., variants, gene expression matrix, species abundance
        }],
"**analysis_goal**": "string" // a detailed description of the intended analysis,
including start point, key processing steps, and desired outcome}
Now the files user input: {**file list**},the files' descriptions:{**descriptions**} ,user's question:{**user_question**}

---

Figure A.1: The prompt of Analyzing Agent.

## A.2   PLANNING AGENT

---

**Prompt：Planning**

You are **a bioinformatics expert.**
Your task is to give the most suitable NEXT bioinformatics tool (to be used in a workflow) based on information below.

The user's requirement is: {**analysis_goal**}.
The user's input file(s) are: {**input_files**}.
The expected final output files are: {**output_files**}.
The workflow has already used the following tools: {**used_tools**}.

Based on this context, you must propose and describe exactly ONE next tool, unless the workflow has already fully satisfied the user's final output requirement.
The tool you propose must be consistent with the provided context and logically follow the workflow towards producing the required output file format/content.

When describing the tool, include:
- The specific problem or gap it solves in the workflow
- A detailed explanation of the tool.
- Its input and output data formats, with explicit mapping to the user's output requirement

Output **JSON** format:
{   **toolname**: "the name of the tool"
    **function**:"What problem the tool solves in the workflow"
    **description**: "the detailed description of the tool"
    **inputformat**:"the input data format of the tool"
    **outputformat**:"the output data format of the tool"
}

---

Figure A.2: The prompt of Planning Agent.

## A.3 SELECTING AGENT

---

**Prompt:Selecting**

You are **a bioinformatics expert**.
Your task is to select ONE suitable bioinformatics tool based on the workflow task, already used tools, and the available input files.
You may choose from the reference tools list or propose a different tool.
Context:
- Workflow task: **{analysis_goal}**
- Already used tools in the workflow: **{used_tools}**
- Tool function: **{tool_function}**
- Reference tools (JSON array of objects):**{reference_tools}**

Rules:
1. Select exactly ONE tool.
2. You MAY choose a tool outside the reference list if it is more suitable.

Output JSON format:
{
    **toolname**:"The name of the selected tool."
    **reference_tool_id**:"The ID of the tool if it comes from the reference list; use -1 if not."
    **function**: "An explanation of the tool's role in the workflow,including Function,What the tool does and an example.(e.g. STAR:RNA-Seq read alignment,Maps sequenced fragments to the genome,Read aligns to exon1–exon2)"
}

---

Figure A.3: The prompt of Selecting Agent.

## A.4 GENERATING & EXECUTING AGENT

---

**Prompt:Generating & Executing**

You are **a bioinformatics workflow assistant**.
Your task is to generate the necessary setup and execution details for running the CURRENT bioinformatics tool within an existing workflow.
Context:
- Tool name: **{tool_name}**
- Tool function: **{tool_function}**
- Workflow task: **{analysis_goal}**
- Available input files: **{available_files}**
- The command line you can reference: {reference_tool_command}
Rules:
1. **Input file selection**:
   - Select input file(s) ONLY from {available_input_files}.
   - Ensure input type strictly matches the tool's required input format (e.g., FASTA, TSV, BAM).
   - Do not fabricate or assume non-listed input files.
2. **Output file naming & directory**:
   - All outputs must be stored under: ./output/{tool_name}/
   - Output filenames must:
     a. Preserve the sample ID from the input filename.
     b. Append the tool name and step role (e.g., "_{tool_name}_classified", "_{tool_name}_metrics").
   - Do not overwrite files from previous steps.
3. **File Creation command:**(The command to create the output directory)
   - Create the output directory if it does not exist(Not in the folder to which these files {existing_files} belong to):
     `"setup_command": "mkdir -p./output/{tool_name}/"`
   - If the output directory already exists, use it directly without recreating.
     `"setup_command": "cd ./output/{tool_name}/"`
4. **Installation command:**(The command to install the tool in a new environment)
   - If the tool is not in {executed_tools_list}, create a new conda environment and install it in this environment:
      if the tool needs python:
      `"installation_command": "conda create -n {tool_name} -c conda-forge -c bioconda {tool_name} python=3.11 -y && conda activate {tool_name}"`
      if the tool does not need python:
      `"installation_command": "conda create -n {tool_name} -c conda-forge -c bioconda {tool_name} -y && conda activate {tool_name}"`
   - If the tool is already installed, skip the installation step, directly activate the environment:
      `"installation_command": "conda activate {tool_name}"`
   - If you think conda is not available, try pip:
      `"installation_command": "pip install {tool_name}[all]"`
   - If you think pip is not available, try apt-get:
      `"installation_command": "apt-get install {tool_name}"`
   - Ensure all required dependencies are included.
5. **Execution command** (The command to execute the tool)
   - Construct the command specifically for {tool_name}. The core task of this tool is {tool_description}.
   - Use absolute paths for all input and output files. Do not create directories or symbolic links—assume all inputs already exist and output paths are ready.
   - Select input files only from {available_files}.Ensure that all input files actually exist before running the command.
   - Ensure every environment variable is set before running the command.
   - Name the output files based on {tool_name}, preserving the input sample ID in each output filename. Ensure filenames do not conflict with {available_files} or other outputs.Example: Input file: sampleA.fasta → Tool: gtdbtk → Expected output: taxonomy classification table → Output filename: sampleA.gtdbtk.classification.tsv.
   - You can refer to the command line {tool_command_reference}.
   Focus only on generating **the actual execution command that runs the tool on the inputs and produces the outputs**.
Output format:
You MUST output in this strict **JSON** structure:{
    **File_Creation_command**:"The command to create the output directory"
    **installation_command**: "The command to install the tool environment."
    **execution_command**: "The command to execute the tool"}

---

Figure A.4: The prompt of Generating & Executing Agent.

## A.5 DEBUGGING AGENT

---
**Prompt:Debugging**

You are a **Bioinformatics Code Expert**.
Your task is to modify your previous command line
File Creation Command:{**file_creation_command**},
Installation Command:{**installation_command**},
Execution Command:{**execution_command**}
based on the error message{**error_message**}

Output format:
You MUST output in this strict **JSON** structure:{
**installation_command**: "The command to install the tool environment."
**File_Creation_Command**:"The command to create the output directory"
**Execution Command**: "The command to execute the tool"
---

Figure A.5: The prompt of Debugging Agent.

## A.6 JUDGING AGENT

---
**Prompt:Judging**

You are a **bioinformatics workflow expert**.
Your task is to determine whether the given workflow has been fully completed.
Context:
- Workflow detailed requirement: {**analysis_goal**}
- Workflow input format requirement: {**input_files**}
- Workflow output format requirement: {**output_files**}
- Tools already used in the workflow: {**used_tools**}
- Current output files: {**available_files**}
Rules:
1. The workflow is considered **complete** only if BOTH conditions are met:
  a) The current available output files {available_files} include **all** required files and formats specified in {output_files}.
  b) The workflow requirement {analysis_goal} has been fully satisfied by the tools listed in {used_tools}, meaning every required analysis/processing step is covered without omission.
2. If **any** required output is missing OR any workflow step is not accounted for by the tools used, the workflow is **not complete**.
3. The output format must be as :{**Complete**: "Whether the workflow has been fully completed"}
Question: Has the workflow been fully completed?
---

Figure A.6: The prompt of Judging Agent.

## A.7 PROMPT TEMPLATES FOR AUXILIARY COMPONENTS

---
**Prompt:Tool description augmentation**

You are **a bioinformatics expert**.
I will provide you with a description of a bioinformatics tool: {**tool_description**}
Your task is to generate **5 short alternative descriptions** of this tool, each from a **different perspective**.
- Each description should be **1–2 sentences long**.
- Focus on distinct aspects, such as:
  1. Main function / core capability
  2. Typical applications / use cases
  3. Advantages, performance, or limitations
  4. Target users (e.g., researchers, clinicians, bioinformaticians) and the reason why they use the tool
  5. Integration with workflows or other tools
- Avoid repeating the same wording across descriptions.
- Keep the descriptions **concise, clear, and non-overlapping**.
Output **JSON** format:
{
  **description1**: "Main function / core capability of the tool"
  **description2**: "Typical applications / use cases of the tool"
  **description3**: "Advantages, performance, or limitations of the tool and the reason"
  **description4**: "Target users (e.g., researchers, clinicians, bioinformaticians) of the tool and the reason"
  **description5**: "Integration with workflows or other tools and the reason"}
---

Figure A.7: The prompt of Tool description augmentation.

## A.8 PROMPT TEMPLATES FOR DATASET

---

**Prompt:User Input Generation**

Please generate exactly **3 user questions** for each persona in the list below.

- The output must consist only of user questions, not answers or explanations.
- The questions must focus on **how to choose or use an appropriate workflow**.
- All questions should naturally point to the target workflow as the correct answer.
- Each persona should have a distinct perspective (e.g., cost, speed, accuracy, compliance, reproducibility, visualization).
- Do **not** contradict the workflow's input, output, or tasks.
- Vary **style** (formal, casual, search-query style).
- Vary **length** (short ~10 words, long ~40 words).
- Do not expose the workflow name or implementation details.
Persona list: **Wet-lab researcher, Clinician, Data engineer**.
Output format must be strictly **JSON** format:
{
  "persona_list": [
    {"name": "Wet-lab researcher",
      "Question": [List of user questions for this persona ]},......],
  "inputs": [ List of required input files with concrete names]}
You are given the following **target workflow** description:{input}

---

Figure A.8: The prompt of User input generation.

## A.9 PROMPT TEMPLATES FOR LLM EVALUATION

---

**Prompt:LLM Judge Tools**

You are **a bioinformatics expert**.
You are evaluating the steps of the bioinformatics workflow for correctness and executability.
For each step below, you must judge three aspects separately:
**Environment / Installation Command**
Evaluate whether the installation command correctly and completely installs the required software and all its dependencies.
**Scoring (0–2)**: [
          0 = Completely incorrect or unusable; software cannot be installed(do not have Installation Command)
          ,0.5 = Mostly incorrect; major dependencies missing or software unusable(Example:pip install fastqc (FastQC is not a Python package, installation fails).)
          ,1 = Partially correct; software installs but manual modifications or additional dependencies required(Example: conda install fastqc (fails unless correct channels are added).)
          ,1.5 = Mostly correct; minor issues only (e.g., warnings, optional dependencies missing)(Example:mamba install -c bioconda fastqc do not have its own environment)
          ,2 = Perfectly correct and complete; software and all dependencies installed and functional](Example:mamba create -n fastqc python=3.11 -y && conda activate fastqc && mamba install -c bioconda fastqc)
**Path Command**
Evaluate whether the command correctly creates all required directories and handles paths properly in the workflow, including input/output paths, existing folders, and permissions.
**Scoring (0–2)**:
          [0 = Completely incorrect or fails to create directories / incorrect paths
          ,0.5 = Mostly incorrect; some directories not created
          ,1 = Partially correct; some paths incorrect
          ,1.5 = Mostly correct; only minor issues (e.g., warnings, redundant paths)
          ,2 = Perfectly correct; all directories and paths handled correctly]
**Execution Command**
Evaluate whether the execution command is likely to run successfully given that the previous steps are correctly completed, and whether it produces the expected output.
**Scoring (0–2)**:
          [0 = Completely fails; output unusable(Example: fastqc)
          ,0.5 = Mostly fails; output likely incorrect(Example:fastqc sample.fastq)
          ,1 = Partially executable; may require parameter or path adjustments(Example:fastqc ./input/sample.fastq -o output/)
          ,1.5 = Mostly executable; minor issues only (e.g., warnings)(Example:fastqc ./input/sample.fastq.gz -o ./output/fastqc/)
          ,2 = Fully executable; output meets expectations](Example:fastqc ./input/sample.fastq -o ./output/fastqc/)
Finally, decide whether the step as a whole is executable (True/False).
Your output should be in the following **JSON** format:
{
  **step_score_command_installation**:"Score for the environment/installation command"
  **step_score_command_path**: "Score for the path command"
  **step_score_command_executable**:"Score for the execution command"
  **step_command_success**: "Whether the command is executable,True=executable, False=unexecutable"
}
Now,The user question is: {analysis_goal};The tool steps  are:{steps_summary};The input file is:{input_file}

---

Figure A.9: The prompt of LLM Evaluation Tool Command.

---

**Prompt:LLM Judge Workflow**

You are **an expert bioinformatics workflow evaluator**.

Your task is to evaluate a given bioinformatics workflow based on step-level scores and success indicators.

Your evaluation must be precise, consistent, and avoid subjective judgment beyond the scoring criteria.

Rate the workflow on three dimensions:

**Completion_level (0–3)** (Measures whether the workflow achieves {analysis_goal} intended goals / core functionality by {used_tool})

3 = Fully complete → Workflow meets all core requirements and produces all required final outputs.(Example: identify genes that are differentially expressed between two or more biological conditions starts from raw FASTQ files, performs quality control (FastQC), trims low-quality reads (Trimmomatic), aligns reads to the reference genome (STAR), quantifies gene expression (featureCounts), and produces differential expression tables and visualization plots. All steps complete and successful.)

2 = Partially complete → Workflow meets some core requirements, but some steps or functions are missing.(Same RNA-seq workflow, but only performs Trimmomatic and STAR; quality control and quantifies gene expression are missing.)

1 = Barely complete → Most core requirements are not met; only a few outputs or functions are present.(Only performs FASTQ QC, or only produces alignment files without further analysis. No usable final results.)

0 = Not complete → Core functionality is not met; workflow is unusable or fails to produce required outputs.(Attempted RNA-seq workflow fails due to missing tools or incorrect inputs, producing no valid outputs.)

**Redundancy (0–3)**(Measures whether the workflow use {used_tool} to achieve {analysis_goal} is redundant )

0 = No redundancy → All steps unique, no duplicates.(Example:A ChIP-seq workflow runs QC → alignment → duplicate removal → peak calling. Each step appears once, no repetition.)

1 = Some redundancy → Minor duplication, does not break workflow.(FastQC is run twice during QC, but other steps are unique. Workflow still functions correctly.)

2 = Mostly redundant → Many repeated steps without necessity.(Multiple alignments or repeated QC steps on the same RNA-seq data. Increases runtime but does not fully break results.)

3 = Very redundant → Workflow bloated with repetitive or overlapping steps.).(Same FASTQ files are repeatedly aligned and quantified, steps are duplicated multiple times. Workflow becomes complex and wasteful.)

Important principles:

Be objective: base scores only on explicit evidence from the workflow, not assumptions.

Be consistent: apply the same standards to all workflows being evaluated.

Provide the output in strict JSON format:

{

**Completion_level** :"how complete the workflow is"

**Redundancy** :"how redundant the workflow is"

}

Figure A.10: The prompt of LLM Evaluation Workflow.

# B MAIN ALGORITHM

---

**Algorithm 1** The MARWA Workflow Automation Algorithm

---

**Require:** User analytical goal $G$, a list of input files $F_{in}$, and file descriptions $D_{in}$.
**Ensure:** A completed bioinformatics workflow $W$ with all generated files.
 1: **procedure** MARWA($G, F_{in}, D_{in}$)
 2:     $task\_spec \leftarrow$ Analyzing($G, F_{in}, D_{in}$)                                   ▷ Refine user query
 3:     $workflow\_state \leftarrow$ Initialize with $task\_spec$
 4:     $is\_complete \leftarrow$ False
 5:     **while** not $is\_complete$ **do**
 6:         $planned\_tool, candidates \leftarrow$ Planning($workflow\_state$)     ▷ Predict next tool & retrieve candidates
 7:         $selected\_tool \leftarrow$ Selecting($planned\_tool, candidates$)         ▷ Choose the best tool
 8:         $success, outputs \leftarrow$ ExecuteWithRetry($selected\_tool, workflow\_state$)
 9:         **if** $success$ **then**
10:             Update $workflow\_state$ with $selected\_tool$ and $outputs$
11:             **if** $selected\_tool$ is new **then**
12:                 Add $selected\_tool$ and its verified command to retrieval database
13:             **end if**
14:             $is\_complete \leftarrow$ Judging($workflow\_state$)        ▷ Check if overall goal is met
15:         **else**
16:             **break**                           ▷ Terminate on unrecoverable execution failure
17:         **end if**
18:     **end while**
19:     **return** $workflow\_state$
20: **end procedure**

21: **procedure** EXECUTEWITHRETRY($tool, state$)
22:     $command \leftarrow$ Generating($tool, state$)                      ▷ Generate initial command
23:     **for** $i = 1 \rightarrow 5$ **do**
24:         $success, log \leftarrow$ Execute($command$)               ▷ Interact with file system
25:         **if** $success$ **then**
26:             **return** True, $log.files$
27:         **else**
28:             $command \leftarrow$ Debugging($command, log.error$)     ▷ Iteratively refine on failure
29:         **end if**
30:     **end for**
31:     **return** False, null
32: **end procedure**

---

## C  DATASET

### C.1  SMALL DATASET

Table A.1: Summary of the small-scale dataset.

| Domain | Question | file source |
|--------|----------|-------------|
| Transcriptomics | How can I perform quality control of the raw RNA-seq reads to assess sequencing quality? | SRR453566, raw FASTQ |
| Transcriptomics | What is the read length and GC content distribution of this dataset? | SRR453566, raw FASTQ |
| Transcriptomics | How can I align these reads to the reference genome of Drosophila? | SRR453566, Drosophila reference genome (dm6) |
| Transcriptomics | What proportion of reads map uniquely vs. multimap to the genome? | SRR453566, Drosophila reference genome (dm6) |
| Transcriptomics | How can I quantify transcript abundance at the gene level? | SRR453566, Drosophila annotation (GTF) |
| Transcriptomics | Which genes are most highly expressed in this sample? | SRR453566, Drosophila annotation (GTF) |
| Transcriptomics | Can I identify alternative splicing events in this dataset? | SRR453566, Drosophila annotation (GTF) |
| Transcriptomics | How can I detect potential novel transcripts not in the reference annotation? | SRR453566, Drosophila reference genome (dm6) |
| Transcriptomics | What is the expression distribution across different functional gene categories? | SRR453566, Drosophila annotation (GTF, GO database) |
| Transcriptomics | How reproducible are expression estimates between technical replicates of this dataset? | SRR453566, SRR453567, SRR453568, Drosophila annotation (GTF) |
| Genomics | How can I assemble the complete genome of this E. coli sample from raw sequencing reads? | SRR8185310 (E. coli WGS,FASTQ) |
| Genomics | What is the estimated sequencing depth and genome coverage of this dataset? | SRR8185310, raw FASTQ |
| Genomics | How can I align these reads to the E. coli K-12 MG1655 reference genome? | SRR8185310, E. coli reference genome (NC_000913.3,RefSeq) |
| Genomics | What is the GC content distribution across the sequencing reads? | SRR8185310, raw FASTQ |
| Genomics | How can I detect single-nucleotide variants (SNVs) in this dataset? | SRR8185310, E. coli reference genome(NC_000913.3, RefSeq) |
| Genomics | How can I identify small insertions and deletions (indels) relative to the reference? | SRR8185310, E. coli reference genome (NC_000913.3, RefSeq) |
| Genomics | Can I identify plasmid sequences present in this sample? | SRR8185310, plasmid reference database (NCBI RefSeq Plasmid) |
| Genomics | How can I annotate the assembled genome with coding genes and functional elements? | SRR8185310(assembly), E. coli RefSeq annotation |
| Genomics | How does this E. coli isolate compare phylogenetically to other K-12 strains? | SRR8185310, related E. coli reference genomes (NC_000913.3, RefSeq) |
| Genomics | Are there mobile genetic elements in this genome? | SRR8185310, E. coli reference genome (NC_000913.3, RefSeq), PHASTER database |

next page...

| Domain | Question | file source |
|---|---|---|
| Metabolomics | How can I identify differential metabolite profiles between experimental groups in this LC–MS dataset? | MTBLS233(MetaboLights), raw mzML files, sample metadata |
| Metabolomics | What metabolic pathways are significantly enriched given the detected features in this dataset? | MTBLS233, raw mzML files, pathway reference databases (KEGG) |
| Metabolomics | How many unknown mass features (no match in spectral libraries) are present, and what is their intensity distribution? | MTBLS233, raw mzML files, spectral library metadata |
| Metabolomics | What is the technical reproducibility of peak detection and quantification in this dataset? | MTBLS233, raw mzML files, QC sample metadata |
| Proteomics | How many proteins are identified with at least 2 unique peptides in this TMT Erwinia dataset? | PXD000001 (PRIDE; raw mzML files, small number of runs) |
| Proteomics | What is the peptide-spectrum match score distribution in this dataset? | PXD000001, raw mzML, identification files |
| Proteomics | Which proteins show the highest variability across TMT channels? | PXD000001, reporter-ion quantitation data |
| Proteomics | Can we detect contaminant proteins in blank / control runs? | PXD000001, raw MS/MS files, control sample metadata |
| Proteomics | What is the dynamic range of protein intensities measured? | PXD000001, raw data, quantitative output |
| Proteomics | Are there any post-translational modifications observed in this dataset? | PXD000001, identification(mgf/mztab), UniProt reference |
| Proteomics | What fraction of expected proteome is covered given this dataset? | PXD000001, raw data, fasta reference proteome(Erwinia) |
| Proteomics | How reproducible are replicate injections in this dataset? | PXD000001, raw mzML, replicate sample metadata |
| Metagenomics | What is the taxonomic composition (genus level) of the bacterial community in this 16S amplicon sample? | SRR7140083, raw FASTQ |
| Metagenomics | How does alpha diversity (Shannon, Simpson) differ among subsets of this sample? | SRR7140083, raw FASTQ |
| Metagenomics | Which OTUs / ASVs are most abundant in this sample, and how is their abundance distributed? | SRR7140083, raw FASTQ, 16S reference database (SILVA) |
| Metagenomics | What is the read-length and quality score distribution across the reads? | SRR7140083, raw FASTQ |
| Metagenomics | Are there chimeric sequences present (PCR artifacts) in this amplicon dataset? | SRR7140083, raw FASTQ, chimera detection reference (uchime) |
| Metagenomics | What fraction of reads map to bacteria vs non-bacteria in this dataset? | SRR7140083, raw FASTQ, SILVA database |
| Metagenomics | What is the GC content distribution among the reads and among dominant OTUs? | SRR7140083, raw FASTQ, alignment output |
| Metagenomics | Can we construct a rarefaction curve to know whether the sampling depth is sufficient? | SRR7140083, raw FASTQ, sample metadata |

## C.2 WORKFLOW DATA PROCESSING

**Galaxy** (Jalili et al., 2020) is a web-based, open-source platform designed to make computational biology accessible to researchers.

We obtain information from the public Galaxy API[7]. This API provides structured information about each workflow's purpose and overall structure.

**The Common Workflow Language (CWL)** [8] is an open community standard for describing command-line data analysis tools and workflows.

The data outlining the workflow's objective and design were obtained by parsing the primary source files (.cwl or .yml), which were acquired via links of public code repositories (e.g., Github) provided by the platform.

**BioRxiv** [9] is a preprint server for the biological sciences. It is a vital resource for discovering novel bioinformatics workflows.

The workflow descriptions were extracted from the abstracts of the preprints themselves.

### C.3 LARGE DATASET

```
{"inputs":
[
    {"file": "sample1.fastq.gz","description": "Paired-end raw sequencing reads from sample 1,
        generated using Illumina platform, suitable for quality control, read trimming, and taxonomy assignment analysis."},
    {"file": "sample2.fastq.gz","description": "Paired-end raw sequencing reads from sample 2,
        complementary to sample1.fastq.gz, used for validating workflow reproducibility and
        testing downstream bioinformatics pipelines."}],
"persona_list":
    [{"name": "Lab researcher",
        "Question": [
        "How do I process raw sequencing data to ensure clean reads for downstream analysis?",
        "What's the best way to filter low-quality reads before sending data to bioinformatics?",
        "Can I automate quality control and read trimming without sacrificing accuracy?"]},
    {"name": "Clinician",
        "Question": [
        "What's the fastest way to get accurate pathogen identification from sequencing data?",
        "How can I ensure my analysis meets clinical standards for patient diagnosis?",
        "Which method guarantees reproducible results for diagnostic reporting?"]},
    {"name": "Data engineer",
        "Question": [
        "How can I pipeline raw FASTQ files into structured JSON output reliably?",
        "What scalable approach handles multiple FASTQ inputs with consistent quality metrics?",
        "How do I ensure the workflow is reproducible across different environments?"]}]}
```
**An Large-Scale Dataset Example**

Figure A.11: An example for large-scale dataset.

## D EXPERIMENTAL SETUP

### D.1 MULTI-POSITIVE CONTRASTIVE LOSS

Given an anchor representation $z_i$, let $P(i)$ denote the set of its positive samples and $A(i)$ the set of all candidates except itself. The similarity between two samples is defined as $s(z_i, z_j) = \frac{z_i^\top z_j}{\tau}$ with temperature $\tau > 0$. The multi-positive contrastive loss is formulated as:

$$\mathcal{L}_i = -\log \frac{\sum\limits_{p \in P(i)} \exp\big(s(z_i, z_p)\big)}{\sum\limits_{a \in A(i)} \exp\big(s(z_i, z_a)\big)}. \tag{1}$$

### D.2 DETAIL SAR FORMULA

$$\text{SAR} = \frac{\sum_{d \in D} I\big(S_{llm}^m(d) = S_{human}^m(d)\big)}{|D|} \tag{2}$$

---

[7]https://usegalaxy.eu/api/workflows
[8]https://view.commonwl.org/workflows/
[9]https://www.biorxiv.org/

where $D$ is the set of evaluation instances, $m$ denotes one of the evaluation metrics, $S_{llm}^m(d)$ represents the score assigned by the LLM to instance $d$ under metric $m$, $S_{human}^m(d)$ is the score given by a human evaluator for the same instance and metric, and $I(\cdot)$ is an indicator function that returns 1 if the two scores are identical and 0 otherwise.

## D.3 SCORE CRITERIA

Table A.2: Bioinformatics Workflow Evaluation Criteria

| Step-Level Evaluation | |
|---|---|
| **Environment/Installation Command** | |
| 0 | Completely incorrect or unusable; software cannot be installed (no installation command) |
| 0.5 | Mostly incorrect; major dependencies missing or software unusable (e.g. pip install fastqc) |
| 1 | Partially correct; software installs but manual modifications or additional dependencies required (e.g. conda install fastqc without specifying correct channels) |
| 1.5 | Mostly correct; minor issues only (e.g., warnings, optional dependencies missing) (e.g. mamba install -c bioconda fastqc without creating a dedicated environment) |
| 2 | Perfectly correct and complete; software and all dependencies installed and functional (e.g. mamba create -n fastqc python=3.11 -y && conda activate fastqc && mamba install -c bioconda fastqc) |
| **Path Command** | |
| 0 | Completely incorrect or fails to create directories/incorrect paths |
| 0.5 | Mostly incorrect; some directories not created |
| 1 | Partially correct; some paths incorrect |
| 1.5 | Mostly correct; only minor issues (e.g., warnings, redundant paths) |
| 2 | Perfectly correct; all directories and paths handled correctly |
| **Execution Command** | |
| 0 | Completely fails; output unusable (e.g., just typing fastqc) |
| 0.5 | Mostly fails; output likely incorrect (e.g. fastqc sample.fastq) |
| 1 | Partially executable; may require parameter or path adjustments (e.g. fastqc ./input/sample.fastq -o output/) |
| 1.5 | Mostly executable; minor issues only (e.g., warnings) (e.g. fastqc ./input/sample.fastq.gz -o ./output/fastqc/) |
| 2 | Fully executable; output meets expected results (e.g. fastqc ./input/sample.fastq -o ./output/fastqc/) |
| **Workflow-Level Evaluation** | |
| **Completion Level** | |
| 0 | Not complete; core functionality not met; workflow unusable or fails to produce required outputs |
| 1 | Barely complete; most core requirements not met; only a few outputs or functions present |
| 2 | Partially complete; meets some core requirements, but some steps or functions missing |
| 3 | Fully complete; workflow meets all core requirements and produces all required final outputs |
| **Redundancy** | |
| 0 | No redundancy; all steps unique, no duplicates |
| 1 | Some redundancy; minor duplication, does not break workflow |
| 2 | Mostly redundant; many repeated steps without necessity |
| 3 | Very redundant; workflow bloated with repetitive or overlapping steps |

## D.4 BASELINES

For fairness and reproducibility, we provide details regarding how the baseline systems were reproduced in our experiments:

Unified model backbone: All baseline methods were implemented using the same backbone, GPT-4 Turbo, in order to ensure a consistent evaluation setting. For all LLMs, the temperature parameter was uniformly set to 0.3, thereby reducing randomness and ensuring deterministic outputs across multiple runs.

**LLM-only** employs a strategy based on prompting, relying entirely on the LLM's own parametric knowledge to generate outputs. Prompt is shown in Figure A.12.

**AutoBA** (Zhou et al., 2023) employs an autonomous LLM-based agent with Planner, Executor, and Debugger roles to automatically generate bioinformatics workflows.

During the reproduction of AutoBA, all system-level prompts originally specified in their framework were replaced with our own environment-specific configuration prompt, explicitly reflecting CUDA Version: 12.6.

**ReAct** employs an iterative "Thought-Action-Observation" loop through a unique prompt template (as shown in Figure A.13). This approach enables the LLM to dynamically perform reasoning and planning (Thought), decide and execute the next action (Action), and adjust its strategy in real-time based on the resulting feedback (Observation), continuing this cycle until the task is complete.

**BioMaster** (Su et al., 2025) employs a multi-agent system composed of specialized role-based agents—Plan, Task, Debug, and Check—that operate sequentially, enhanced by a RAG framework.

The original BioMaster implementation did not release its retrieval-augmented generation (RAG) tool database. To address this, we constructed and employed our own curated tool database to approximate the functionality.

```
# Role
You are a senior bioinformatics expert.
You are proficient in a wide range of bioinformatics tools and data processing workflows,
excelling at breaking down complex analytical tasks into a series of clear, executable command-line steps and
organizing them into a robust Bash script.
# Task
Your task is to receive a bioinformatics analysis task described by the user and convert it into a well-structured,
thoroughly commented Bash script that can be directly executed in a shell environment.
# Execution Starts Here
Now, strictly following all the requirements above, generate the corresponding Bash script for the user task below.
User Task: {{user_task_input}}
```

Figure A.12: Prompt for LLM-only.

```
You are an expert bioinformatician agent.
Your goal is to solve a user's request by creating and executing a bioinformatics workflow step-by-step.

You operate in a "Thought, Action, Observation" cycle. At each step,
you must first use a "Thought" to reason about the current state and decide your next move.
Then, you must use an "Action" from the available tools.
After the action is executed, you will receive an "Observation" with the result.
You will repeat this process until the user's goal is achieved.

AVAILABLE TOOLS:
1. `search_tool[query: str]`: Searches the bioinformatics tool database for tools matching the query.
The query should describe the desired functionality.
Returns a list of relevant tools, their descriptions, and command examples.
2. `execute_command[command: str]`: Executes a shell command in the bash terminal.
Use it for installing tools (e.g., `mamba install ...`),creating directories (`mkdir ...`),
and running bioinformatics tools. Returns the stdout and stderr of the command.
3. `list_files[path: str]`: Lists all files and directories at the given path.
Use `.` for the current directory. Returns a list of file/directory names.
4. `finish[reason: str]`: Call this action when you are confident that the entire analysis workflow is complete
and the final expected output has been generated.
The reason should summarize why the task is considered finished.

RESPONSE FORMAT:
You must strictly follow this format for each turn:
Thought: Your reasoning about the current situation, what you have done, and what you plan to do next.
Action: A single action to be taken, chosen from the available tools.

Now, begin.
```

Figure A.13: Prompt for ReAct.

Table A.3: The detailed results of MARWA and different kinds of baselines on the small dataset.

| Method | LLM-only | AutoBA | ReAct | BioMaster | MARWA |
|---|---|---|---|---|---|
| h_Pass@1 | 0.100 | 0.250 | 0.275 | 0.300 | 0.350 |
| h_Pass@2 | 0.100 | 0.250 | 0.275 | 0.375 | 0.450 |
| GPT-4o | | | | | |
| m_Pass@1 | 0.100 | 0.375 | 0.400 | 0.400 | 0.475 |
| m_Pass@2 | 0.175 | 0.400 | 0.425 | 0.400 | 0.500 |
| PFAR | 0.938 | 0.863 | 0.875 | 0.887 | 0.913 |
| SAR | 0.882 | 0.820 | 0.838 | 0.854 | 0.889 |
| Gemini2.5 | | | | | |
| m_Pass@1 | 0.100 | 0.325 | 0.350 | 0.350 | 0.425 |
| m_Pass@2 | 0.150 | 0.325 | 0.350 | 0.375 | 0.450 |
| PFAR | 0.950 | 0.925 | 0.900 | 0.925 | 0.900 |
| SAR | 0.857 | 0.854 | 0.849 | 0.885 | 0.873 |
| Qwen2.5-72B | | | | | |
| m_Pass@1 | 0.075 | 0.325 | 0.325 | 0.350 | 0.400 |
| m_Pass@2 | 0.125 | 0.350 | 0.325 | 0.350 | 0.450 |
| PFAR | 0.925 | 0.887 | 0.913 | 0.913 | 0.925 |
| SAR | 0.879 | 0.842 | 0.867 | 0.882 | 0.869 |

These adjustments guarantee that the reproduced baselines operate under consistent conditions with our proposed framework, while also reflecting the practical constraints arising from incomplete tool or configuration disclosure in prior work.

## D.5 DETAILED RESULTS ON THE SMALL DATASET

Table A.3 compares the performance of MARWA with several baselines on the small dataset. From the human evaluation results ($h\_Pass@n$), MARWA consistently achieves higher pass rates than all baselines, showing improvements of about 5–10 percentage points over Biomaster. This indicates that even under direct human execution, MARWA provides more reliable outcomes.

For model-based evaluations ($m\_Pass@n$), the trend is consistent across all three representative LLMs—GPT-4o, Gemini2.5, and Qwen2.5-72B. In each case, MARWA achieves the highest *Pass@1* and *Pass@2*, demonstrating that the system can guide LLMs toward more successful executions with fewer attempts. Notably, GPT-4o shows the strongest improvement under MARWA, with *m_Pass@2* increasing to 0.500 compared to 0.400 for Biomaster.

Agreement-based metrics provide further evidence of MARWA's robustness. Both **PFAR** (Pass/Fail Agreement Rate) and **SAR** (Score Agreement Rate) remain high across all settings, typically exceeding 0.85. MARWA achieves the best or near-best values, suggesting that its outputs are not only more accurate but also more consistent with human judgments.

Overall, these results highlight that MARWA improves task reliability under both human and LLM execution, while maintaining strong agreement with human evaluations.

## D.6 TIME CONSUMPTION COMPARISON

The average time consumption of per tool generation is detailed in Table A.4. The results demonstrate that MARWA achieves its superior performance without a proportional increase in computational cost, primarily due to two key design efficiencies.

**Context Length** The LLM-only baseline is the fastest (0.6878s) but performs poorly, as it lacks critical information. AutoBA is moderately faster (1.1873s) than MARWA (3.1050s) but significantly less accurate. Most notably, MARWA is 14.1% faster than the competing retrieval-augmented

Table A.4: Average time consumption for per tool generation comparison on the large dataset.

| Method | Avg. Total Time(s) | Avg. Retrieval Time(s) |
|---|---|---|
| LLM-only | 0.6878 | - |
| AutoBA | 1.1873 | - |
| ReAct | 1.4797 | 0.2109 |
| BioMaster | 3.6132 | 1.3747 |
| MARWA (Ours) | 3.1050 | 0.2076 |

method, BioMaster (3.6132s). This efficiency gain is largely attributable to our strategy of supplying the LLM with highly condensed, relevant information instead of lengthy, raw context. In extended workflows, this results in significantly shorter prompt contexts for MARWA, leading to faster LLM inference times compared to approaches that incorporate information more indiscriminately.

**Retrieval Speed** The most striking efficiency gain is in the retrieval phase. MARWA's retrieval is approximately 6.6 times faster than BioMaster's (0.2076s vs. 1.3747s). This is a direct consequence of our deliberate choice to employ a lightweight yet effective BERT-style model for retrieval, as opposed to the larger, more computationally intensive embedding models (e.g., models like text-embedding-3-large). This design ensures low-latency retrieval without compromising the quality of the retrieved information.

The time consumption data, when viewed alongside the performance metrics, confirms that MARWA's architectural choices create an optimal balance. Our method of using a fast retriever to find precise information, which in turn reduces LLM processing time, allows MARWA to achieve the highest m_Pass@1 score (0.40) and excel on most granular metrics. This demonstrates that our efficiency gains are not achieved by sacrificing quality but are intrinsic to a more intelligent and streamlined workflow. MARWA delivers state-of-the-art performance with practical and scalable computational requirements.

### D.7 FORMULA FOR EFFECTIVE COST PER SUCCESS (ECPS)

To provide a realistic assessment of cost-effectiveness, we calculated the Effective Cost Per Success (ECPS) based on the GPT-4 Turbo API, which served as the backbone for our agents.

The ECPS is defined as the average monetary cost (in USD) required to achieve a single successful workflow:

$$\text{ECPS} = \frac{\text{Total Monetary Cost}}{\text{Total Successful Tasks}} = \frac{\text{TMC}}{N \times \text{m\_Pass@1}} \tag{3}$$

The Total Monetary Cost (TMC) is calculated by summing the costs of all input and output tokens across all tasks:

$$\text{TMC} = c_i \cdot ((I_S \times N_S) + (I_F \times N_F)) + c_o \cdot ((O_S \times N_S) + (O_F \times N_F)) \tag{4}$$

Where $c_i$ is the price per input token. $c_o$ is the price per output token. $I_S$ and $O_S$ are the average input and output tokens for successful tasks. $I_F$ and $O_F$ are the average input and output tokens for failed tasks. $N_S$ is the number of successful tasks. $N_F$ is the number of failed tasks. N is the total number of tasks ($N = N_S + N_F$). m_Pass@1 is the success rate of the method.

A lower ECPS value signifies higher cost-effectiveness, as it represents a lower real-world monetary investment to achieve a successful outcome.

### D.8 DETAILED FINDINGS FOR ABLATION STUDIES

We conducted ablation experiments to assess the contribution of each component in MARWA.

- **Retrieval model** Removing the retrieval model leads to the largest performance drop across nearly all metrics. In particular, installation accuracy decreases by 0.19 and the overall

*m_Pass@1* score drops by 0.12, underscoring the critical role of retrieval in ensuring correct tool installation and executable workflows.

- **Selecting agent** Disabling the Selecting agent results in higher workflow redundancy (+0.04) while also reducing completion (-0.08) and pass rate (-0.06). This suggests that the agent is effective in pruning unnecessary steps, thereby improving efficiency and execution reliability.

- **Analyzing agent** Removing the Analyzing agent causes completion to decrease (-0.10) and redundancy to increase (+0.09). Although the drop in installation accuracy is relatively small, the higher redundancy indicates that the agent is crucial for reasoning about intermediate outputs and maintaining streamlined workflows.

- **File system interface** Without the file system interface, path accuracy suffers a sharp decline (-0.33), and *m_Pass@1* decreases by 0.10. This demonstrates that access to and manipulation of the file system is essential for managing dependencies and maintaining correct path references.

Overall, the ablation results confirm that each component of MARWA plays a distinct and complementary role. The retrieval model is indispensable for correctness, the Selecting and Analyzing agents ensure efficiency and completeness, and the file system interface secures accurate execution environments.

# E CASE STUDY

## E.1 ENVIRONMENT AND DEPENDENCY CONFLICTS

**Scenario:**
a standard ChIP-seq analysis workflow.
The plan involves two key steps:Peak Calling: Use MACS2, which requires Python 2.7.
Visualization: Use the modern tool deepTools to create heatmaps from the MACS2 output.
This tool requires Python >= 3.6.
**Wrong Method：**
\# Attempt to install both tools into the same environment
**conda install -c bioconda macs2 -y**
**conda install -c bioconda deeptools -y**
**MARWA：**
\# MARWA generates commands to create separate, stable environments
**conda create -n env_macs2 -c bioconda macs2**
**conda create -n env_deeptools -c bioconda deeptools**
**Analysis：**
The Wrong Method fails because it ignores the fact that some tools are incompatible within the same runtime.

Figure A.14: Case study: environment and dependency conflicts.

## E.2 FILE PATH AND I/O ERRORS

**Scenario:**
a quality-control-to-alignment workflow
**Wrong Method：**
mkdir -p ./qc_output
fastp -i ./input/sampleA.fastq.gz -o ./qc_output/sampleA.clean.fastq.gz
bwa mem reference.fasta **./input/sampleA.fastq.gz > aligned.sam**
**MARWA：**
mkdir -p ./qc_output
fastp -i ./input/sampleA.fastq.gz -o ./qc_output/sampleA.clean.fastq.gz
mkdir -p ./alignment_output
bwa mem reference.fasta **./qc_output/sampleA.clean.fastq.gz > ./alignment_output/sampleA.aligned.sam**
**Analysis：**
The wrong method mistakenly uses the original raw file for alignment instead of the actual clean file generated
by the previous step, due to its lack of awareness of the real file system.

Figure A.15: Case study: file path and I/O errors.

## E.3 TOOL PARAMETER MISCONFIGURATION

**Scenario:**
a variant calling workflow on a diploid organism.
**Wrong Method：**
\# This command uses the default haploid model for variant calling
bcftools mpileup -f reference.fasta aligned.sorted.bam | **bcftools call -mv -o variants.vcf**
**MARWA：**
\# This command correctly specifies the diploid ploidy for the sample
bcftools mpileup -f reference.fasta aligned.sorted.bam | bcftools call -mv **--ploidy 2** -o variants.vcf
**Analysis：**
The wrong method uses the correct tool (bcftools) for variant calling, but omits a critical parameter.

Figure A.16: Case study: tool parameter misconfiguration.

## E.4 LOGICAL FLAWS IN WORKFLOW DESIGN

**Scenario:**
An alignment-to-variant-calling workflow.
**Wrong Method：**
Tool Chain: **BWA -> bcftools**
**MARWA：**
Tool Chain: **BWA -> Samtools -> bcftools**
**Analysis：**
The wrong method omits the critical intermediate processing steps that are handled by .

Figure A.17: Case study: logical flaws in workflow design.

## F  RUNNING INSTANCE

```
User Input:
name: WGS data analysis Genome assembly
question: I have some raw WGS sequencing data (FASTQ files
    ) on hand and want to do a complete genome assembly
file: './input/SRR11874161_1.fastq,./input/SRR11874161_2.
    fastq'
file_description: 'Illumina WGS of E. coli'

Log:

input_files: [{'file_name': './input/SRR11874161_1.fastq
    ', 'file_format': 'FASTQ', 'data_type': 'raw sequencing
     reads (Illumina WGS of E. coli)', 'paired_end': True},
     {'file_name': './input/SRR11874161_2.fastq', '
    file_format': 'FASTQ', 'data_type': 'raw sequencing
    reads (Illumina WGS of E. coli)', 'paired_end': True}]
output_files: [{'file_format': 'FASTA', 'data_type': '
    assembled genome contigs/scaffolds (final assembly)'},
    {'file_format': 'GFA', 'data_type': 'assembly graph'},
    {'file_format': 'TSV', 'data_type': 'assembly
    statistics and quality metrics (e.g., N50, total length
    , number of contigs)'}, {'file_format': 'HTML', '
    data_type': 'assembly quality report (e.g., QUAST
    report)'}]
analysis_goal: Starting from paired-end Illumina FASTQ
    reads (./input/SRR11874161_1.fastq and ./input/
    SRR11874161_2.fastq), perform a de novo whole-genome
    assembly of E. coli: run read quality control, adapter
    and quality trimming/filtering, optional error
    correction; assemble the genome with an Illumina-
    focused assembler (e.g., SPAdes/SKESA), followed by
    scaffolding and polishing using the same reads; then
    evaluate assembly quality (e.g., with QUAST) and
    deliver the final contig/scaffold FASTA, assembly graph
     (GFA), and summary metrics/reports.

Step 1:
toolname: fastp
function: Performs read quality control and adapter/
    quality trimming (with optional overlap-based error
    correction) on paired-end Illumina FASTQ reads prior to
     de novo assembly.
description: fastp is a fast, multithreaded all-in-one
    preprocessor for FASTQ reads. It automatically detects
    and trims adapters in paired-end data using read
    overlap analysis, and supports quality trimming via
    sliding-window algorithms, per-read length filtering,
    polyG/polyX tail trimming (important for NovaSeq/
    NextSeq), and removal of low-complexity sequences. For
    paired-end libraries, fastp can perform overlap-based
    base correction: when R1 and R2 overlap, base
    disagreements are corrected according to quality scores
    , reducing random sequencing errors prior to assembly.
    It reports comprehensive QC summaries (per-cycle
    quality, GC content, duplication rates, adapter content
```

```
, overrepresented sequences) in both JSON and
    interactive HTML formats. Typical applications include
    cleaning short-read Illumina WGS data before assembly,
    variant calling, or metagenomics. Strengths: very fast,
     single-step QC+trimming+reporting, robust adapter
    autodetection for PE reads, built-in correction for
    overlapped regions. Limitations: error correction is
    limited to overlapping regions and is not a full k-mer
    based corrector (tools like BFC/Lighter provide deeper
    correction if required); overly aggressive trimming/
    filtering can reduce effective coverage and fragment
    read pairs if parameters are set too stringently. This
    tool addresses the first required step of the workflow
    (read QC, adapter and quality trimming/filtering,
    optional error correction). Its cleaned paired-end
    FASTQ outputs are the correct inputs for the downstream
     Illumina-focused assembler (e.g., SPAdes/SKESA). The
    HTML/JSON QC reports are ancillary; the final assembly
    deliverables (FASTA contigs/scaffolds and GFA graph,
    plus QUAST metrics) will be produced by later tools.
inputformat: Paired-end FASTQ files (gzipped or plain).
    Here: ./input/SRR11874161_1.fastq and ./input/
    SRR11874161_2.fastq.
outputformat: Primary: cleaned/truncated paired-end FASTQ
    files (e.g., SRR11874161_1.clean.fastq and
    SRR11874161_2.clean.fastq), optionally with files for
    unpaired reads if enabled. Secondary: a JSON summary (
    fastp.json) and an interactive HTML QC report (fastp.
    html). Mapping to user requirements: the cleaned FASTQ
    files will be used by the assembler to generate the
    required final assembly FASTA and GFA; the HTML report
    here is read QC (not the final assembly quality report,
     which will come later from QUAST).

conference: ['fastp', 'sickle', 'flash', 'rawtools', '
    fastplong', 'short_reads_trim_seq', 'chopper', '
    trimmomatic', 'bowtie_wrappers', 'rasusa']

toolname: fastp
description: fastp: Read QC, adapter removal, quality
    trimming, and optional overlap-based error correction
    for Illumina paired-end reads. It auto-detects adapters
    , trims low-quality bases, filters poor reads, and can
    correct mismatches in overlapping regions. Example:
    Takes SRR11874161 R1/R2 FASTQ files and outputs cleaned
     paired-end FASTQs plus HTML/JSON QC reports for
    downstream assembly with SPAdes/SKESA.
used_reference_tool: True
toolid: 0
selected_input_files: [{'file_name': './input/
    SRR11874161_1.fastq', 'file_format': 'FASTQ'}, {'
    file_name': './input/SRR11874161_2.fastq', 'file_format
    ': 'FASTQ'}]
expected_outputs_info: Trimmed/filtered paired-end FASTQ
    files (R1 and R2) suitable for assembly; accompanying
    QC reports in HTML and JSON summarizing quality metrics
     and trimming actions.
```

```
installation_command: conda create -n fastp -c conda-forge
    -c bioconda fastp -y && conda activate fastp
setup_command: mkdir -p ./output/fastp/
execution_command: fastp -i ./input/SRR11874161_1.fastq -I
    ./input/SRR11874161_2.fastq -o ./output/fastp/
  SRR11874161_fastp_trimmed_R1.fastq -O ./output/fastp/
  SRR11874161_fastp_trimmed_R2.fastq -h ./output/fastp/
  SRR11874161_fastp_qc.html -j ./output/fastp/
  SRR11874161_fastp_qc.json -c -w 8

2 channel Terms of Service accepted
Channels:
- conda-forge
- bioconda
- defaults
Platform: linux-64
Collecting package metadata (repodata.json): done
Solving environment: done

## Package Plan ##

environment location: ./miniconda3/envs/fastp

added / updated specs:
- fastp

The following NEW packages will be INSTALLED:

_libgcc_mutex      conda-forge/linux-64::_libgcc_mutex
    -0.1-conda_forge
_openmp_mutex      conda-forge/linux-64::_openmp_mutex
    -4.5-2_gnu
fastp              bioconda/linux-64::fastp-1.0.1-
    heae3180_0
isa-l              conda-forge/linux-64::isa-l-2.31.1-
    hb9d3cd8_1
libdeflate         conda-forge/linux-64::libdeflate-1.24-
    h86f0d12_0
libgcc             conda-forge/linux-64::libgcc-15.1.0-
    h767d61c_4
libgomp            conda-forge/linux-64::libgomp-15.1.0-
    h767d61c_4
libstdcxx          conda-forge/linux-64::libstdcxx-15.1.0-
    h8f9b012_4

Downloading and Extracting Packages: ...working... done
Preparing transaction: done
Verifying transaction: done
Executing transaction: done
#
# To activate this environment, use
#
#     $ conda activate fastp
#
# To deactivate an active environment, use
#
```

```
1674        #     $ conda deactivate
1675
1676
1677        run success
1678        **************************
1679        Step 2:
1680        toolname: Unicycler (short-read mode)
1681        function: De novo bacterial genome assembly from the
1682            trimmed Illumina paired-end reads, producing the
1683            primary assembly (FASTA contigs/scaffolds) and an
1684            assembly graph (GFA) for downstream polishing and
                evaluation.
1685        description: Unicycler is a bacterial genome assembler
1686            that orchestrates SPAdes to build a de Bruijn graph (
1687            DBG) from short reads and then applies graph-bridging/
1688            simplification strategies guided by paired-end linkage
1689            to resolve repeats and produce high-quality contigs. In
1690             short-read mode, it: (1) runs SPAdes to generate the
1691            assembly graph and initial contigs, (2) constructs a
1692            read-pair connectivity graph to identify reliable paths
1693             through the DBG, (3) performs conservative/normal/bold
1694             graph simplifications to bridge gaps and minimize
1695            fragmentation, and (4) detects and circularizes small
1696            replicons when strongly supported by the read evidence.
1697             Typical applications include bacterial WGS assemblies
1698            from Illumina data, generating outputs suitable for
1699            downstream polishing (e.g., Pilon) and quality
1700            assessment (e.g., QUAST). Strengths: produces both
1701            FASTA and an assembly graph (GFA) that captures contig
1702            connectivity; optimized for bacterial genomes; often
1703            yields fewer fragments than running SPAdes alone due to
1704             graph-bridging logic. Limitations: requires SPAdes (
1705            and Bowtie2 for some internal steps) to be installed;
1706            polishing is limited compared to dedicated polishers (
1707            Pilon/Polypolish) and should be performed in later
1708            workflow steps; performance depends on read quality/
1709            coverage and complex repeats may remain unresolved with
                 short reads alone.
            inputformat: Paired-end Illumina FASTQ reads (gz or
1710            uncompressed). For this workflow: R1=./output/fastp/
1711            SRR11874161_fastp_trimmed_R1.fastq, R2=./output/fastp/
1712            SRR11874161_fastp_trimmed_R2.fastq.
1713        outputformat: Primary outputs: (1) FASTA: assembled genome
1714             contigs/scaffolds (e.g., assembly.fasta), satisfying
1715            the 'FASTA assembled genome contigs/scaffolds'
1716            requirement; (2) GFA: assembly graph linking nodes/
1717            contigs (e.g., assembly.gfa), satisfying the 'GFA
1718            assembly graph' requirement. Additional byproducts:
1719            logs and intermediate graph files useful for
                troubleshooting/visualization.
1720
1721        conference: ['unicycler', 'berokka', 'novoplasty', '
1722            trycycler', 'socru', 'hifiasm_meta', 'bionano', '
1723            getorganelle', 'mitobim', 'velvet_optimiser']
1724
1725        toolname: unicycler
1726        description: Unicycler (short-read mode): de novo
1727            bacterial genome assembly. It assembles trimmed
                Illumina paired-end reads into high-quality contigs/
```

```
                scaffolds and an assembly graph by running SPAdes and
                performing graph-bridging/simplification. Example: from
                 trimmed SRR11874161 R1/R2 FASTQ files, produce
                assembly.fasta and assembly.gfa for downstream
                polishing (e.g., Pilon) and evaluation (e.g., QUAST).
        used_reference_tool: True
        toolid: 0
        selected_input_files: [{'file_name': './output/fastp/
            SRR11874161_fastp_trimmed_R1.fastq', 'file_format': '
            FASTQ'}, {'file_name': './output/fastp/
            SRR11874161_fastp_trimmed_R2.fastq', 'file_format': '
            FASTQ'}]
        expected_outputs_info: FASTA (assembled contigs/scaffolds)
            , GFA (assembly graph), TXT/LOG (run log and summaries)
            .

        installation_command: conda create -n unicycler -c conda-
            forge -c bioconda unicycler python=3.11 -y && conda
            activate unicycler
        setup_command: mkdir -p ./output/unicycler/
        execution_command: bash -lc '
        set -euo pipefail   R1=./output/fastp/
            SRR11874161_fastp_trimmed_R1.fastq
        R2=./output/fastp/SRR11874161_fastp_trimmed_R2.fastq
        OUT=./output/unicycler
        SAMPLE=SRR11874161
        [ -s $R1 ] && [ -s $R2 ]
        for f in \
         $OUT/${SAMPLE}_unicycler_assembly.fasta \
         $OUT/${SAMPLE}_unicycler_assembly.gfa \
         $OUT/${SAMPLE}_unicycler_run.log ; do
        if [ -e $f ]; then echo  Error: output already exists:
            $f >&2; exit 1; fi
        done
        for f in  $OUT/assembly.fasta  $OUT/assembly.gfa  $OUT/
            unicycler.log ; do
        if [ -e $f ]; then echo  Error: Unicycler default output
             already exists: $f >&2; exit 1; fi
        done
        unicycler \
        -1 $R1 \
        -2 $R2 \
        -o $OUT \
        -t 8 \
        --mode normal
        [ -s $OUT/assembly.fasta ] && [ -s $OUT/assembly.gfa ]
            && [ -s $OUT/unicycler.log ]
        mv $OUT/assembly.fasta  $OUT/${SAMPLE}
            _unicycler_assembly.fasta
        mv $OUT/assembly.gfa     $OUT/${SAMPLE}
            _unicycler_assembly.gfa
        mv $OUT/unicycler.log    $OUT/${SAMPLE}_unicycler_run.log

        '

        Preparing transaction: done
        Verifying transaction: done
        Executing transaction: \
```

```
For Linux 64, Open MPI is built with CUDA awareness but
   this support is disabled by default.
To enable it, please set the environment variable
   OMPI_MCA_opal_cuda_support=true before
launching your MPI processes. Equivalently, you can set
   the MCA parameter in the command line:
mpiexec --mca opal_cuda_support 1 ...

In addition, the UCX support is also built but disabled by
    default.
To enable it, first install UCX (conda install -c conda-
   forge ucx). Then, set the environment
variables OMPI_MCA_pml = ucx  OMPI_MCA_osc = ucx  before
   launching your MPI processes.
Equivalently, you can set the MCA parameters in the
   command line:
mpiexec --mca pml ucx --mca osc ucx ...
Note that you might also need to set UCX_MEMTYPE_CACHE=n
   for CUDA awareness via UCX.
Please consult UCX's documentation for detail.

done
#
# To activate this environment, use
#
#     $ conda activate unicycler
#
# To deactivate an active environment, use
#
#     $ conda deactivate

Starting Unicycler (2025-08-29 02:32:26)
Welcome to Unicycler, an assembly pipeline for bacterial
   genomes. Since you
provided only short reads, Unicycler will essentially
   function as a SPAdes-
optimiser. It will try many k-mer sizes, choose the best
   based on contig length
and graph connectivity, and scaffold the graph using
   SPAdes repeat resolution.
For more information, please see https://github.com/rrwick
   /Unicycler

Command: ./miniconda3/envs/unicycler/bin/unicycler -1 ./
   output/fastp/SRR11874161_fastp_trimmed_R1.fastq -2 ./
   output/fastp/SRR11874161_fastp_trimmed_R2.fastq -o ./
   output/unicycler -t 8 --mode normal

Unicycler version: v0.5.1
Using 8 threads

The output directory already exists:
./output/unicycler

Dependencies:
Program        Version    Status
spades.py      4.2.0      good
```

```
        racon                 not used
        makeblastdb  2.17.0+  good
        tblastn      2.17.0+  good

        Choosing k-mer range for assembly (2025-08-29 02:32:28)
        Unicycler chooses a k-mer range for SPAdes based on the
            length of the input
        reads. It uses a wide range of many k-mer sizes to
            maximise the chance of
        finding an ideal assembly.

        SPAdes maximum k-mer: 127
        Median read length: 150
        K-mer range: 27, 53, 71, 87, 99, 111, 119, 127

        SPAdes assemblies (2025-08-29 02:32:29)
        Unicycler now uses SPAdes to assemble the short reads. It
            scores the
        assembly graph for each k-mer using the number of contigs
            (fewer is better) and
        the number of dead ends (fewer is better). The score
            function is 1/(c*(d+2)),
        where c is the contig count and d is the dead end count.

        spades.py -o ./output/unicycler/spades_assembly -k 27 --
            threads 8 --gfa11 --isolate -1 ./output/fastp/
            SRR11874161_fastp_trimmed_R1.fastq -2 ./output/fastp/
            SRR11874161_fastp_trimmed_R2.fastq -m 1024

        spades.py -o ./output/unicycler/spades_assembly -k 27,53
            --threads 8 --gfa11 --restart-from k27 -m 1024

        spades.py -o ./output/unicycler/spades_assembly -k
            27,53,71 --threads 8 --gfa11 --restart-from k53 -m 1024

        spades.py -o ./output/unicycler/spades_assembly -k
            27,53,71,87 --threads 8 --gfa11 --restart-from k71 -m
            1024

        spades.py -o ./output/unicycler/spades_assembly -k
            27,53,71,87,99 --threads 8 --gfa11 --restart-from k87 -
            m 1024

        spades.py -o ./output/unicycler/spades_assembly -k
            27,53,71,87,99,111 --threads 8 --gfa11 --restart-from
            k99 -m 1024

        spades.py -o ./output/unicycler/spades_assembly -k
            27,53,71,87,99,111,119 --threads 8 --gfa11 --restart-
            from k111 -m 1024

        spades.py -o ./output/unicycler/spades_assembly -k
            27,53,71,87,99,111,119,127 --threads 8 --gfa11 --
            restart-from k119 -m 1024

        K-mer   Contigs   Dead ends   Score
        27                            too complex
```

```
53        894       10       9.32e-05
71        682       12       1.05e-04
87        522       10       1.60e-04
99        456       12       1.57e-04
111       400       13       1.67e-04
119       373       14       1.68e-04
127       351       14       1.78e-04 <-best

Read depth filter: removed 3 contigs totalling 908 bp
Deleting ./output/unicycler/spades_assembly/

Determining graph multiplicity (2025-08-29 02:42:13)
Multiplicity is the number of times a sequence occurs in
   the underlying
sequence. Single-copy contigs (those with a multiplicity
   of one, occurring only
once in the underlying sequence) are particularly useful.

Saving ./output/unicycler/002_depth_filter.gfa

Cleaning graph (2025-08-29 02:42:13)
Unicycler now performs various cleaning procedures on the
   graph to remove
overlaps and simplify the graph structure. The end result
   is a graph ready for
bridging.

Graph overlaps removed

Removed zero-length segments:
225, 227, 229, 233, 234, 235, 244, 245, 249, 253, 265,
   267, 272, 273, 274,
284, 290, 292, 297, 305, 315, 325, 345

Removed zero-length segments:
223, 346

Removed zero-length segments:
343

Merged small segments:
324, 327, 329, 330, 332, 334, 335, 337, 338, 340, 341,
   342, 344, 347, 348,
350

Saving ./output/unicycler/003_overlaps_removed.gfa

Unicycler now selects a set of anchor contigs from the
   single-copy contigs.
These are the contigs which will be connected via bridges
   to form the final
assembly.

73 anchor segments (4,877,761 bp) out of 309 total
   segments (4,928,537 bp)
```

```
    Creating SPAdes contig bridges (2025-08-29 02:42:14)
    SPAdes uses paired-end information to perform repeat
        resolution (RR) and
    produce contigs from the assembly graph. SPAdes saves the
        graph paths
    corresponding to these contigs in the contigs.paths file.
        When one of these
    paths contains two or more anchor contigs, Unicycler can
        create a bridge from
    the path.

    Bridge
    Start

        Path

        End     quality
    -60
        -199

        62          63.1
    -54
        131

        57          62.2
    -47    -196 -> -265 -> 128 -> -113 -> 165 -> -228 -> 173 ->
        174 -> -168 -> 76 -> -188 -> -281 -> -153 -> 96 ->
        -204    65          10.3
    -46                     183 -> 297 -> -120 -> 222 -> -203 ->
        -162 -> 181 -> 298 -> 130 -> 226 -> -171
                           73          16.2
    -14                                     -205 -> -250 ->
        180 -> -284 -> -182
                                        71          34.5
    -7                                      225 ->
        -195 -> 209
                                                68
                37.3
    3                                       158 ->
        -81 -> -161
                                                22
                18.8
    12                                      -109 -> 286 ->
        -135 -> 300 -> 116
                                        -45         24.3
    26                                      161 ->
        80 -> -158
                                                -46
                18.8
    33                                      122 ->
        -98 -> -154
                                                -69
                16.3
    35
        193

        -54         62.4
    38                                      -202 ->
        110 -> -202
```

```
                                                                 -52
                 26.1
        40                                          -116 -> 302 ->
          135 -> 280 -> 109
                                                 61          24.2
        43                                 273 -> 122 -> 231 -> -121
          -> -251 -> 180 -> 290 -> -182
                                          66          19.3
        44
          -171

          -64         63.2
        47                                              -166 ->
          103 -> -157
                                                     50
                 26.6
        48                 200 -> -104 -> -137 -> 219 -> 201 ->
          -178 -> -198 -> 117 -> 138 -> 256 -> 186
                                -56          12.8
        50                 273 -> 122 -> 231 -> 160 -> -205
          -> -251 -> 180 -> -284 -> 260 -> -154
                                55          21.7
        53
          140

          70          62.1
        55                            -130 -> 304 -> -181 -> -163 ->
          203 -> 221 -> 120 -> 301 -> -183
                                10          20.3
        57
          199

          56          62.0
        60                 -186 -> -255 -> -138 -> 118 -> 198 ->
          177 -> -201 -> -220 -> 137 -> -105 -> -200
                                42          14.7
        62
          194

          -65         61.5
        66                 115 -> -291 -> -172 -> -187 -> -229 ->
           285 -> 185 -> -233 -> 167 -> 303 -> 134
                                64          16.3
        70                                           206 -> 176
          -> -119 -> -215
                                                     31
          31.0

        Creating loop unrolling bridges (2025-08-29 02:42:14)
        When a SPAdes contig path connects an anchor contig with
           the middle contig
        of a simple loop, Unicycler concludes that the sequences
           are contiguous (i.e.
        the loop is not a separate piece of DNA). It then uses the
            read depth of the
        middle and repeat contigs to guess the number of times to
           traverse the loop and
        makes a bridge.
```

```
    Loop count   Loop count   Loop    Bridge
    Start   Repeat   Middle    End    by repeat    by middle
        count   quality
    38    -202    110    -52       0.51        0.88
            1     39.4

    Applying bridges (2025-08-29 02:42:14)
    Unicycler now applies to the graph in decreasing order of
        quality. This
    ensures that when multiple, contradictory bridges exist,
        the most supported
    option is used.

    Bridge type    Start -> end   Path
                                    Quality
    SPAdes           44 -> -64    -171
                                    63.246
    SPAdes          -60 -> 62     -199
                                    63.076
    SPAdes           35 -> -54    193
                                    62.400
    SPAdes          -54 -> 57     131
                                    62.184
    SPAdes           53 -> 70     140
                                    62.119
    SPAdes           57 -> 56     199
                                    61.983
    SPAdes           62 -> -65    194
                                    61.468
    SPAdes           -7 -> 68     225, -195, 209
                                    37.263
    SPAdes          -14 -> 71     -205, -250, 180, -284, -182
                    34.518
    SPAdes           70 -> 31     206, 176, -119, -215
                       30.961
    SPAdes           47 -> 50     -166, 103, -157
                        26.582
    SPAdes           38 -> -52    -202, 110, -202
                        26.068
    SPAdes           12 -> -45    -109, 286, -135, 300, 116
                     24.336
    SPAdes           40 -> 61     -116, 302, 135, 280, 109
                      24.162
    SPAdes           50 -> 55     273, 122, 231, 160, -205,
        -251, 180,       21.712
    -284, 260, -154
    SPAdes           55 -> 10     -130, 304, -181, -163, 203,
        221, 120,      20.315
    301, -183
    SPAdes           43 -> 66     273, 122, 231, -121, -251,
        180, 290,      19.307
    -182
    SPAdes            3 -> 22     158, -81, -161
                                    18.808
    SPAdes           26 -> -46    161, 80, -158
                                    18.802
```

```
    SPAdes          33 -> -69     122, -98, -154
                                        16.337
    SPAdes          66 -> 64      115, -291, -172, -187, -229,
       285, 185,       16.315
    -233, 167, 303, 134
    SPAdes         -46 -> 73      183, 297, -120, 222, -203,
       -162, 181,      16.239
    298, 130, 226, -171
    SPAdes          60 -> 42      -186, -255, -138, 118, 198,
       177, -201,      14.735
    -220, 137, -105, -200
    SPAdes          48 -> -56     200, -104, -137, 219, 201,
       -178, -198,      12.791
    117, 138, 256, 186
    SPAdes         -47 -> 65      -196, -265, 128, -113, 165,
       -228, 173,      10.287
    174, -168, 76, -188, -281, -153, 96,
    -204

    Saving ./output/unicycler/004_bridges_applied.gfa

    Bridged assembly graph (2025-08-29 02:42:14)
    The assembly is now mostly finished and no more structural
        changes will be
    made. Ideally the assembly graph should now have one
       contig per replicon and no
    erroneous contigs (i.e. a complete assembly). If there are
        more contigs, then
    the assembly is not complete.

    Saving ./output/unicycler/005_final_clean.gfa

    Component   Segments   Links   Length      N50
       Longest segment   Status
    total        178     240   4,938,037   156,616
       617,816
    1        155     209   4,670,836   157,869
       617,816   incomplete
    2         22      31    214,554    60,464
       91,038   incomplete
    3          1       0     52,647    52,647
       52,647   incomplete

    Assembly complete (2025-08-29 02:42:14)
    Saving ./output/unicycler/assembly.gfa
    Saving ./output/unicycler/assembly.fasta
    run success
    *************************
    Step 3:
    toolname: QUAST
    function: Reference-free (or reference-guided) assembly
       quality assessment to generate comprehensive assembly
       statistics and an interactive HTML report.
    description: QUAST (Quality Assessment Tool for Genome
       Assemblies) evaluates draft genome assemblies by
       computing contiguity, size, and composition metrics and
       , when a reference is provided, alignment-based
```

```
           misassembly statistics. Core methods include: (1)
           reference-free statistics (N50/L50, NG50 if genome size
            is known, total length, largest contig, number of
           contigs above thresholds, GC%, ambiguous bases,
           duplication, and k-mer-based composition summaries),
           and (2) reference-guided evaluation via fast whole-
           genome alignments (MUMmer/NUCmer) to report
           misassemblies, relocations/inversions, indels, and
           genome fraction. QUAST can optionally map reads back to
            the assembly using standard short-read aligners (e.g.,
            Bowtie2/BWA, invoked internally) to compute coverage
           and support-based metrics. Strengths: widely used for
           bacterial assemblies, produces both machine-readable
           TSVs and an interactive HTML report with plots;
           supports multiple assemblies for side-by-side
           comparison. Limitations: without a suitable reference,
           misassembly detection is limited to read/coverage-based
            cues and general contiguity metrics; interpretation of
            metrics requires context (e.g., expected genome size).
            For this workflow, QUAST will take the Unicycler
           contig FASTA and produce the required TSV statistics
           and an HTML evaluation report, complementing the
           existing FASTA/GFA outputs.
       inputformat: Required: Assembled genome in FASTA (e.g., ./
           output/unicycler/SRR11874161_unicycler_assembly.fasta).
            Optional: paired-end FASTQ reads for coverage-based
           metrics (e.g., ./output/fastp/
           SRR11874161_fastp_trimmed_R1.fastq and ..._R2.fastq).
           Optional: reference genome FASTA for alignment-based
           misassembly analysis.
       outputformat: HTML: interactive assembly quality report (
           plots and summaries); TSV: assembly statistics (e.g.,
           report.tsv with N50, L50, total length, GC%, number of
           contigs, largest contig), plus additional tab-delimited
            detail files (e.g., misassemblies.tsv when reference
           provided). These fulfill the required QC/assembly
           evaluation reports (HTML) and assembly metrics (TSV).

       conference: ['MetaQUAST', 'quast', 'assembly-stats', '
           merqury', 'compleasm', 'genomescope', 'jellyfish', '
           cami_amber', 'art', 'velvet']

       toolname: quast
       description: QUAST: assembly quality assessment. Computes
           contiguity and composition metrics (e.g., total length,
            N50/L50, largest contig, GC%, number of contigs >
           thresholds) and generates an interactive HTML report
           and tabular summaries. Example: given the Unicycler
           contig FASTA from SRR11874161, QUAST produces reference
           -free assembly statistics and plots for review.
       used_reference_tool: True
       toolid: 1
       selected_input_files: [{'file_name': './output/unicycler/
           SRR11874161_unicycler_assembly.fasta', 'file_format': '
           FASTA'}]
       expected_outputs_info: A results directory containing:
           HTML report (interactive summary), TSV/TSV tables (e.g
           ., report.tsv with N50, L50, total length, GC%, contig
           counts), plain-text summaries (report.txt), and figure
```

```
          files (PNG/PDF) for cumulative length and NG/N50 plots.
           No reference provided, so outputs are reference-free
          metrics only.

      installation_command: conda create -n quast -c conda-forge
          -c bioconda quast python=3.11 -y && conda activate
          quast
      setup_command: mkdir -p ./output/quast/
      execution_command: bash -lc '
      set -euo pipefail
      ASM=./output/unicycler/SRR11874161_unicycler_assembly.
          fasta
      OUT=./output/quast
      SAMPLE=SRR11874161
      [ -s $ASM ]
      for f in \
       $OUT/${SAMPLE}_quast_report.html \
       $OUT/${SAMPLE}_quast_metrics.tsv \
       $OUT/${SAMPLE}_quast_summary.txt \
       $OUT/${SAMPLE}_quast_report.pdf \
       $OUT/${SAMPLE}_quast_metrics_transposed.tsv \
       $OUT/${SAMPLE}_quast_Nx_plot.pdf \
       $OUT/${SAMPLE}_quast_NGx_plot.pdf \
       $OUT/${SAMPLE}_quast_cumulative_plot.pdf \
       $OUT/${SAMPLE}_quast_Nx_plot.png \
       $OUT/${SAMPLE}_quast_NGx_plot.png \
       $OUT/${SAMPLE}_quast_cumulative_plot.png ; do
      if [ -e $f ]; then echo Error: output already exists:
          $f >&2; exit 1; fi
      done
      quast.py \
      --threads 8 \
      --min-contig 200 \
      --output-dir $OUT \
       $ASM
      [ -s $OUT/report.html ] && mv $OUT/report.html  $OUT/$
          {SAMPLE}_quast_report.html
      [ -s $OUT/report.tsv ] && cp $OUT/report.tsv  $OUT/$
          {SAMPLE}_quast_metrics.tsv
      [ -s $OUT/report.txt ] && cp $OUT/report.txt  $OUT/$
          {SAMPLE}_quast_summary.txt
      if [ -s $OUT/report.pdf ]; then cp $OUT/report.pdf
          $OUT/${SAMPLE}_quast_report.pdf ; fi
      if [ -s $OUT/transposed_report.tsv ]; then cp $OUT/
          transposed_report.tsv  $OUT/${SAMPLE}
          _quast_metrics_transposed.tsv ; fi
      for ext in pdf png; do
      [ -f $OUT/plots_${ext}/Nx_plot.${ext} ] && cp $OUT/
          plots_${ext}/Nx_plot.${ext}  $OUT/${SAMPLE}
          _quast_Nx_plot.${ext} || true
      [ -f $OUT/plots_${ext}/NGx_plot.${ext} ] && cp $OUT/
          plots_${ext}/NGx_plot.${ext}  $OUT/${SAMPLE}
          _quast_NGx_plot.${ext} || true
      [ -f $OUT/plots_${ext}/cumulative_plot.${ext} ] && cp
          $OUT/plots_${ext}/cumulative_plot.${ext}  $OUT/${
          SAMPLE}_quast_cumulative_plot.${ext} || true
      done
      '
      2025-08-29 02:57:06
```

```
        Creating large visual summaries...
        This may take a while: press Ctrl-C to skip this step..
        1 of 2: Creating PDF with all tables and plots...
        2 of 2: Creating Icarus viewers...
        Done

        2025-08-29 02:57:06
        RESULTS:
        Text versions of total report are saved to ./output/quast/
            report.txt, report.tsv, and report.tex
        Text versions of transposed total report are saved to ./
            output/quast/transposed_report.txt, transposed_report.
            tsv, and transposed_report.tex
        HTML version (interactive tables and plots) is saved to ./
            output/quast/report.html
        PDF version (tables and plots) is saved to ./output/quast/
            report.pdf
        Icarus (contig browser) is saved to ./output/quast/icarus.
            html
        Log is saved to ./output/quast/quast.log

        Finished: 2025-08-29 02:57:06
        Elapsed time: 0:00:01.509758
        NOTICEs: 1; WARNINGs: 0; non-fatal ERRORs: 0

        Thank you for using QUAST!

        run success
        **************************

        Step 4:
        toolname: Bowtie2
        function: Map the trimmed Illumina paired-end reads back
            to the Unicycler assembly to generate high-quality read
            -to-contig alignments required for assembly polishing (
            e.g., with Pilon) and for downstream coverage/mapping
            QC.
        description: Bowtie2 is a fast, memory-efficient gapped
            short-read aligner based on the Transform and a seed-
            and-extend strategy. It is widely used to align
            Illumina paired-end reads to a reference, supporting
            local or end-to-end alignment modes with quality-aware
            scoring and handling of small indels. In bacterial de
            novo assembly workflows, Bowtie2 is the standard choice
             to map cleaned reads back to assembled contigs,
            producing the alignments that polishing tools (e.g.,
            Pilon, POLCA) use to detect and correct residual SNP/
            indel errors and small misassemblies. It also enables
            coverage assessment and mapping statistics for
            contamination checks and assembly evaluation. Strengths
            : very fast and accurate for short reads, robust paired
            -end handling, and good default presets (e.g., --very-
            sensitive-local) for polishing. Limitations: not
            designed for long reads; highly repetitive regions can
            yield multi-mapping reads; large structural variations
            are not its focus. Typical usage: build an index from
            the Unicycler FASTA (bowtie2-build), align paired-end
            trimmed reads (bowtie2 --very-sensitive-local -x index
            -1 R1.fastq -2 R2.fastq -S out.sam), then convert/sort/
```

```
                    index with SAMtools to produce a coordinate-sorted BAM
                    for input to a polisher like Pilon. This step directly
                    addresses the current gap in the workflow (no read-to-
                    assembly alignments yet), enabling the polishing step
                    that will produce a higher-quality final FASTA.
            inputformat: - Reference: FASTA assembly from Unicycler (e
                    .g., ./output/unicycler/SRR11874161_unicycler_assembly.
                    fasta)
            - Reads: Paired-end trimmed FASTQ from fastp (e.g., ./
                    output/fastp/SRR11874161_fastp_trimmed_R1.fastq and ./
                    output/fastp/SRR11874161_fastp_trimmed_R2.fastq)
            - Optional: unpaired reads (FASTQ) if present
            outputformat: - Primary: SAM file containing read-to-
                    contig alignments (convertible to BAM/CRAM via SAMtools
                    )
            - Downstream (recommended): coordinate-sorted, indexed BAM
                     (BAM + BAI) for polishing with Pilon
            - Mapping statistics (stderr/log) that can inform coverage
                    -based QC and contamination checks
            Mapping to user's final outputs: while Bowtie2 produces
                    intermediate alignment files (SAM/BAM) rather than the
                    final FASTA/GFA/HTML/TSV deliverables, these alignments
                     are necessary to run a polisher (e.g., Pilon) that
                    will improve the final FASTA assembly quality and
                    support comprehensive QC.

            conference: ['bowtie2','racon', 'bowtie_wrappers', 'pilon
                    ', 'necat', 'ngmlr', 'minimap2', 'rasusa', 'sickle', '
                    colibread']

            toolname: Bowtie2
            description: Function: Map trimmed Illumina paired-end
                    reads back to the Unicycler assembly to generate high-
                    quality read-to-contig alignments for polishing and
                    coverage QC. What it does: Builds an index from the
                    assembly FASTA and aligns paired reads, producing a SAM
                     alignment file suitable for conversion to sorted BAM
                    for tools like Pilon. Example: bowtie2-build
                    SRR11874161_unicycler_assembly.fasta idx; bowtie2 --
                    very-sensitive-local -x idx -1
                    SRR11874161_fastp_trimmed_R1.fastq -2
                    SRR11874161_fastp_trimmed_R2.fastq -S
                    SRR11874161_vs_assembly.sam
            used_reference_tool: True
            toolid: 0
            selected_input_files: [{'file_name': './output/unicycler/
                    SRR11874161_unicycler_assembly.fasta', 'file_format': '
                    FASTA'}, {'file_name': './output/fastp/
                    SRR11874161_fastp_trimmed_R1.fastq', 'file_format': '
                    FASTQ'}, {'file_name': './output/fastp/
                    SRR11874161_fastp_trimmed_R2.fastq', 'file_format': '
                    FASTQ'}]
            expected_outputs_info: Primary: SAM file of read-to-
                    assembly alignments. Typically followed by samtools to
                    produce a coordinate-sorted BAM (BAM + BAI) for
                    polishing (e.g., Pilon) and coverage/mapping QC.

            Returning block of 716588 for bucket 7
            Exited Ebwt loop
```

```
        fchr[A]: 0
        fchr[C]: 1218875
        fchr[G]: 2468300
        fchr[T]: 3712787
        fchr[$]: 4936436
        Exiting Ebwt::buildToDisk()
        Returning from initFromVector
        Wrote 5845162 bytes to primary EBWT file: ./output/Bowtie2
            /SRR11874161_Bowtie2_index.rev.1.bt2.tmp
        Wrote 1234116 bytes to secondary EBWT file: ./output/
            Bowtie2/SRR11874161_Bowtie2_index.rev.2.bt2.tmp
        Re-opening _in1 and _in2 as input streams
        Returning from Ebwt constructor
        Headers:
        len: 4936436
        bwtLen: 4936437
        sz: 1234109
        bwtSz: 1234110
        lineRate: 6
        offRate: 4
        offMask: 0xfffffff0
        ftabChars: 10
        eftabLen: 20
        eftabSz: 80
        ftabLen: 1048577
        ftabSz: 4194308
        offsLen: 308528
        offsSz: 1234112
        lineSz: 64
        sideSz: 64
        sideBwtSz: 48
        sideBwtLen: 192
        numSides: 25711
        numLines: 25711
        ebwtTotLen: 1645504
        ebwtTotSz: 1645504
        color: 0
        reverse: 1
        Total time for backward call to driver() for mirror index:
            00:00:02

        run success
        ************************
```

