# OpenReview forum: "MARWA: Multi-agent retrieval-augmented framework for reliable bioinformatics workflow automation"
_ICLR.cc/2026/Conference — Submitted to ICLR 2026_

### Official Review · Reviewer_CtBg · 2025-10-31

**Soundness:** 2
**Presentation:** 3
**Contribution:** 3
**Rating:** 4
**Confidence:** 4

**Summary:**

The paper proposes a multi-agent retrieval-augmented framework call MARWA for reliable bioinformatics Workflow Automation. MARWA's architecture is composed of six specialized LLM-based agents (Analyzing, Planning, Selecting, Generating & Executing, Debugging, and Judging) that operate in a step-by-step, closed-loop fashion. The authors also design an embedding method called LAFT, based on contrastive learning fine-tuning on the pretrained BERT model. Experiments show that MARWA consistently outperforms baselines like AutoBA and BioMaster, particularly in generating correct installation commands and file paths, leading to higher workflow success rates.

**Strengths:**

S1.The six-agent, step-by-step framework is a well-reasoned and significant improvement over one-shot generation. By breaking the complex problem of workflow creation into discrete and verifiable stages, the system introduces robustness and error-handling capabilities that are critical for this domain.
S2.Experiments show that MARWA and LAFT outperform other baseline methods.

**Weaknesses:**

W1. The paper emphasizes that the proposed method is specifically designed for bioinformatics workflow automation. However, although the evaluation datasets are related to bioinformatics, the architectures of the proposed MARWA and LAFT methods do not appear to have domain-specific optimizations for bioinformatics. Could the authors leverage the characteristic structures of bioinformatics data to optimize the model framework itself (rather than only the prompts)?
W2. In the embedding section, fine-tuning using contrastive learning is already a well-established approach for training embedding models. This work merely uses LLM-generated synthetic data to fine-tune the embedding model(BERT), without introducing a novel method. In addition, state-of-the-art embedding models are often based on decoder-only architectures with larger parameter scales, such as BGE-EN-ICL and Qwen3 Embedding.
W3. The experiments show that MARWA achieves a higher success rate compared to baseline methods. However, given its highly complex workflow structure (including six agents and a loop structure), it is expected to consume significantly more tokens and inference time than other methods. The experiments, however, do not evaluate MARWA’s token costs, inference time, or similar metrics.

**Questions:**

1. Based on Table 1, can the authors compare retrieval performance when replacing the BERT model with more recent open-source base embedding models, such as BGE-EN-ICL or Qwen3 Embedding?
2. Can the authors design experiments to evaluate the cost of MARWA, such as the number of tokens consumed and the average inference time?
3. Can the authors compare MARWA with some general-purpose agent methods, such as ReAct?

---

> ### Author Response · Authors · 2025-11-18
>
> Dear Reviewer,
>
> Thank you for these helpful comments. Our detailed answers are provided below.
>
> > Q1:Based on Table 1, can the authors compare retrieval performance when replacing the BERT model with more recent open-source base embedding models, such as BGE-EN-ICL or Qwen3 Embedding?
>
> Thank you for your valuable question.
>
> Following your suggestion, we have supplemented our paper with a performance comparison against advanced models such as BGE-EN-ICL and Qwen3 Embedding in **Section3.2.1, Table 1**.
>
> | Model                        | Dim  | MRR   | Hit@1 | Hit@3 | Hit@10 |
> |------------------------------|------|-------|-------|-------|--------|
> | Qwen3-Embedding-4B           | 2560 | 0.6065| 0.5226| 0.6382| 0.6884 |
> | bge-en-icl                   | 4096 | 0.6114| 0.5141| 0.6338| 0.7183 |
> | Qwen3-Embedding-8B           | 4096 | 0.6458| 0.5593| 0.6893| 0.7458 |
> | **LAFT**                     | 768  | **0.6686**| **0.5779**| **0.6985**| **0.7638** |
>
> 1.  **Performance Comparison**: As shown in the updated Table 1, our **LAFT method**, which is fine-tuned on BERT, **outperforms** all the models on all key retrieval metrics. This demonstrates the effectiveness of our proposed framework, which combines LLM-based augmentation with contrastive learning.
> 2.  **Reason for Fine-tuning BERT**: We chose BERT as our base model for reasons of **practicality and cost-effectiveness**. The field of bioinformatics is rapidly evolving, with tools and databases being constantly updated. Fine-tuning a lightweight model  allows us to retrain the model with **lower computational costs** and in **less time**. This indicates that our method is not only effective but also **scalability and practical value**, making it well-suited to the rapidly iterating nature of the bioinformatics field.
>
> > Q2: Can the authors design experiments to evaluate the cost of MARWA, such as the number of tokens consumed and the average inference time?
>
> We sincerely thank you for this valuable suggestion.
>
> In the revised manuscript, we have added a **Cost-Effectiveness Analysis section (Section 4.6)** that directly addresses the requested metrics: **token consumption** and **monetary cost per successful workflow**.
>
> 1. **Token consumption**
>    We report average input and output tokens separately for **successful (S)** and **failed (F)** tasks across all methods (measured on the large-scale dataset of 2,270 tasks using GPT-4 Turbo as the unified backbone).
>    MARWA consumes **5,117 input, 1,831 output tokens** on successful tasks — higher than simpler baselines (e.g., LLM-only or AutoBA) because it maintains richer multi-agent context. However, this moderate increase is largely offset by its higher success rate.
>
> 2. **Effective Cost Per Success (ECPS)**
>    To provide a practical metric, we introduce **ECPS** (U.S. dollars spent on average to obtain one fully successful workflow), calculated using the official GPT-4 Turbo pricing.
>    Results are shown below:
>
> | Method    | I-Tokens (S) | O-Tokens (S) | I-Tokens (F) | O-Tokens (F) | ECPS  |
> | --------- | ------------ | :----------- | ------------ | ------------ | ----- |
> | LLM-only  | 298          | 689          | 1396         | 2836         | 0.825 |
> | AutoBA    | 945          | 1043         | 1373         | 2218         | 0.383 |
> | ReAct     | 4085         | 1329         | 6526         | 1963         | 0.521 |
> | BioMaster | 6972         | 1920         | 9203         | 3221         | 0.693 |
> | MARWA     | 5117         | 1831         | 6529         | 2357         | **0.310** |
>
> The results demonstrate that **MARWA** achieves the lowest cost of all methods, with an **ECPS of just $0.310**.
>
> In summary, the newly added cost analysis demonstrates that **MARWA not only outperforms all baselines in accuracy and reliability, but also delivers the lowest real-world monetary cost per successful bioinformatics workflow**.

---

> > ### Author Response · Authors · 2025-11-18
> >
> > > Q3: Can the authors compare MARWA with some general-purpose agent methods, such as ReAct?
> >
> > Thank you for your valuable suggestion.
> >
> > We have added a comparison with the general-purpose method ReAct to test the necessity of our specialized framework. The detailed experimental setup is provided in the **Appendix D.4**.
> >
> > Here is a summary of the new experimental results and our analysis of the performance gap:
> >
> > ### Summary of Experimental Results with ReAct
> >
> > 1. **Small-Scale Dataset Performance (Section 4.4, Table 3)**
> >
> > On the 40 expert-curated tasks, ReAct achieved a manual execution pass rate (h_Pass@1) of **27.5%**. It falls significantly short of **MARWA's 37.5%** pass rate.
> >
> > 2. **Large-Scale Dataset Performance (Section 4.5, Table 4)**
> >
> > On the large-scale benchmark of 2,270 tasks, ReAct's average pass rate (m_Pass@1) was **22%**, again lagging behind **MARWA's 40%**.
> >
> >
> > 3. **Cost-Effectiveness(Section 4.6, Table 5)**
> >
> > Due to ReAct's lower success rate and iterative trial-and-error nature, ReAct is less cost-effective. Its Effective Cost Per Success (ECPS) is **$0.521**, which is higher than **MARWA's $0.310**.
> >
> > ### Analysis
> >
> > MARWA addresses the limitations of ReAct in the following four aspects. A selection of case studies are provided in the **Appendix E**.
> >
> > 1.  **Environmental Dependencies**
> >     *   **ReAct:** Tries to install everything into **one single environment** as it goes, frequently causing a mess of version conflicts and errors.
> >     *   **MARWA:** Creates a **clean, separate environment** specifically for the task *before* execution. This prevents conflicts and guarantees a stable, reliable foundation.
> > 2.  **Inaccurate Tool Retrieval**
> >     *   **ReAct:** Uses a generic search that often gets confused by tools with similar names or functions, leading it to pick the wrong one.
> >     *   **MARWA:** Uses a specialized, fine-tuned search that understands the differences between tools, allowing it to precisely identify the most appropriate one for the task.
> > 3.  **File Path Error**
> >     *   **ReAct:** Often "forgets" or skips checking what files are available and just guesses the path, leading to constant "file not found" errors.
> >     *   **MARWA:** Makes it a mandatory rule: **always check the directory first**.
> > 4.  **Lack of Controllable Step Monitoring**
> >     *   **ReAct:** Can get stuck in an infinite loop when it fails, repeatedly trying the same failed approach and wasting a lot of time and money.
> >     *   **MARWA:** Has built-in controls to prevent loops. It either solves the problem or stops quickly, making it far more efficient and cost-effective.

---

> ### Author Response · Authors · 2025-11-18
>
> > W1: The paper emphasizes that the proposed method is specifically designed for bioinformatics workflow automation. However, although the evaluation datasets are related to bioinformatics, the architectures of the proposed MARWA and LAFT methods do not appear to have domain-specific optimizations for bioinformatics. Could the authors leverage the characteristic structures of bioinformatics data to optimize the model framework itself (rather than only the prompts)?
>
> Thank you for this insightful question.
>
> You are correct that MARWA's high-level architecture (e.g., multi-agent, RAG) appears to be general-purpose on the surface. However, in practice, **every key component of our framework was specifically designed to address the unique "structural challenges" of bioinformatics workflows.** These optimizations are embedded at the **architectural level**, not merely within the prompts.
>
> Specifically, bioinformatics analysis has three typical structural characteristics, and MARWA's architecture is made to address them:
>
> 1. **Sequential Dependency and Fragility (Pipeline Structure):** A bioinformatics analysis is a **step-by-step pipeline**, where a failure or even a minor error in an early step can cause the entire downstream process to collapse.
>    - **Our Architectural Solution:** We deliberately avoided a "one-shot" generation approach and instead designed a **step-by-step generation, execution, and validation loop.** This architecture mimics the real-world workflow of a bioinformatician: "run a step, check the output, then decide the next step." This is an **architectural-level optimization** specifically created to handle the sequential nature of bioinformatics analyses.
> 2. **Environmental State-Dependency (File-based State):** Bioinformatics tool commands are **critically dependent on real file paths**, yet Large Language Models are inherently unaware of the external execution environment. This is one of the most common points of failure for automated workflows.
>    - **Our Architectural Solution:** We integrated a **file system interface** into the framework. This is not just a prompt but a **core architectural component** that allows the agent to "see" the actual files and paths in the environment, enabling it to generate correct and executable commands.
> 3. **Tool Semantic Ambiguity (Tool Heterogeneity):** The bioinformatics field has numerous tools with similar functions but different usages. Relying solely on text descriptions is often insufficient for precise selection.
>    - **Our Architectural Solution:** Our LAFT method **fine-tunes the embedding model itself** through **contrastive learning**. We use an LLM to generate augmented descriptions containing **structured information**, such as "input/output file types" and "typical position in a workflow." This forces the model to learn the fine-grained distinctions between tools, making it a **model-level optimization** tailored for the bioinformatics tool ecosystem.
>
> Your suggestion is highly insightful, and we completely agree that we can further leverage the structural characteristics of bioinformatics data and workflows to optimize the model framework. We plan to build a domain-specific Knowledge Graph (KG) using the widely-accepted **EDAM ontology** as its backbone. EDAM provides a standardized vocabulary for bioinformatics operations, data types, data formats, and topics. In this KG, each tool would be a node, with its inputs, outputs, and functions precisely linked via EDAM terms.
>
>
>
> Thank you again for your great efforts and valuable comments. If you have any further questions or require additional clarification, we welcome continued discussion and engagement.

---

### Official Review · Reviewer_AmHM · 2025-11-01

**Soundness:** 2
**Presentation:** 2
**Contribution:** 2
**Rating:** 4
**Confidence:** 3

**Summary:**

This paper tackles the challenge of **automating bioinformatics workflows**, where existing LLM-based systems often fail due to **ambiguous task definitions, heterogeneous tools, and unreliable one-shot generation**.

To address these issues, the authors propose **MARWA**, a multi-agent retrieval-augmented framework that **decomposes workflow construction into modular stages with dedicated agents** for task clarification, tool retrieval, command synthesis, and error correction.

MARWA enhances tool selection through contrastive-learning-based retrieval and validates reliability via a two-stage evaluation combining expert execution and large-scale LLM-based assessment.

Experiments demonstrate that MARWA substantially improves workflow accuracy, robustness, and scalability over existing baselines.

**Strengths:**

The paper designs a comprehensive end-to-end execution pipeline that covers the entire process，including task analysis, tool selection and workflow execution and validation.

This holistic design ensures not only that each stage is logically grounded and traceable, but also that potential errors can be detected and corrected through contextual feedback, significantly enhancing overall reliability.

**Weaknesses:**

The paper mainly proposes solutions to surface-level problems without uncovering the deeper reasoning gaps between the task requirements and the chosen methods.

For instance, it claims that replacing a one-shot generation process with a step-by-step approach improves workflow reliability, yet it never explains *why* one-shot generation fails in this task setting, *what specific reasoning capabilities* are lacking, or *why* step-by-step reasoning would inherently address them.

Since step-by-step generation is already a standard configuration for large language models, this modification only strengthens a weak baseline rather than constituting a principled innovation.

Moreover, the framework appears as an incremental extension or a more fine-grained reactive pipeline, without demonstrating true multi-agent cooperation or emergent division of labor. Consequently, the proposed contributions lack clear conceptual novelty and task-specific motivation, resulting in a framework that feels more like a layered adaptation of existing paradigms than a genuinely innovative approach.

**Questions:**

1. Have previous studies already introduced benchmark datasets or evaluation protocols for workflow automation, and how were these used to assess reliability or execution quality?
2. What specific improvements or novelties do the proposed dataset and evaluation metrics in this paper provide beyond existing ones — in terms of coverage, realism, or reproducibility?
3. Has the paper reported the computational or financial cost (e.g., model inference time, agent coordination overhead, or GPU usage) associated with the multi-agent setup, and how does this compare to the baselines?

---

> ### Author Response · Authors · 2025-11-18
>
> Dear Reviewer,
>
> Thank you for these helpful comments. Our detailed answers are provided below.
>
> > Q1:Have previous studies already introduced benchmark datasets or evaluation protocols for workflow automation, and how were these used to assess reliability or execution quality?
>
> Thank you for your question. You are correct that previous studies have had their own evaluation methods. However, they suffer from significant limitations in assessing the end-to-end **reliability** and **execution quality** of workflows, which is precisely what motivated us to propose the MARWA evaluation framework. The main shortcomings of previous evaluation methods are:
>
> 1. **Traditional Tool Recommendation Systems:** Their evaluations focused on the **accuracy of next-tool prediction** rather than assessing whether the entire generated workflow is executable or logically sound.
> 2. **Early LLM-based Automation Works:** Their evaluations often relied on small-scale, bespoke task sets and mainly used a **binary "pass/fail" metric**. This approach cannot measure workflow quality (e.g., redundancy) or diagnose the specific causes of failure, such as path or parameter errors.
>
> MARWA's evaluation framework addresses these shortcomings in three key ways:
>
> 1. **Systematic, Multi-Scale Benchmark Datasets:** We constructed a small-scale dataset (40 tasks) for expert manual execution and a large-scale dataset (2,270 tasks) for scalable testing.
> 2. **Multi-Dimensional Evaluation Metrics:** We moved beyond a simple pass/fail metric, designing six indicators to assess **execution quality**.
> 3. **An Innovative Two-Stage Evaluation Protocol:** First, we **validated the reliability of LLMs as evaluators** on the small dataset by demonstrating high agreement with human experts (PFAR, SAR). We then applied this trustworthy LLM-based evaluation method to the large-scale dataset to enable rigorous and scalable benchmarking.
>
> In summary, our framework is the first to provide a **structured, validated, and scalable** solution for assessing the reliability and quality of automated bioinformatics workflows, thereby setting a more rigorous benchmark for the field.
>
> > Q2: What specific improvements or novelties do the proposed dataset and evaluation metrics in this paper provide beyond existing ones — in terms of coverage, realism, or reproducibility?
>
> Thank you for your question. Our work introduces significant improvements in realism, coverage, and reproducibility by advancing both **the dataset and the evaluation scheme** used for benchmarking, addressing key limitations of existing approaches.
>
> Regarding the **dataset**,  we developed a small, expert-scored set of 40 fully executable tasks provides a verifiable "gold standard" and a large-scale benchmark of 2,270 tasks captures a wide diversity of user queries.
>
> For the **evaluation scheme**, we shifted away from simple pass/fail judgments to a multi-dimensional scoring system. This approach allows us to precisely diagnose why a model fails, which is critical for guiding improvements. To objectively assess the capabilities of different models, we adopted a two-stage evaluation framework. First, we validate our LLM-based evaluators against a human-executed gold standard to ensure their reliability. Then, we apply multiple different LLMs as cross-verifying judges for the large-scale benchmark, measuring their inter-rater consistency to ensure objective results.
>
> > Q3: Has the paper reported the computational or financial cost (e.g., model inference time, agent coordination overhead, or GPU usage) associated with the multi-agent setup, and how does this compare to the baselines?
>
> Thank you for raising this critical question.
>
> We have added a **Cost-Effectiveness Analysis (Section 4.6, Table 5)** and calculated the **Effective Cost Per Success (ECPS)**, which is the average dollar cost for one successful workflow.
>
> | Method    | I-Tokens (S) | O-Tokens (S) | I-Tokens (F) | O-Tokens (F) | ECPS  |
> | --------- | ------------ | :----------- | ------------ | ------------ | ----- |
> | LLM-only  | 298          | 689          | 1396         | 2836         | 0.825 |
> | AutoBA    | 945          | 1043         | 1373         | 2218         | 0.383 |
> | ReAct     | 4085         | 1329         | 6526         | 1963         | 0.521 |
> | BioMaster | 6972         | 1920         | 9203         | 3221         | 0.693 |
> | MARWA     | 5117         | 1831         | 6529         | 2357         | 0.310 |
>
> The results demonstrate that **MARWA** achieves the lowest cost of all methods, with an **ECPS of just $0.310**. While MARWA's multi-agent process may appear more complex than a single-shot approach, it is **significantly more resource-efficient in practice**.
>
>
>
> Thank you again for your great efforts and valuable comments. If you have any further questions or require additional clarification, we welcome continued discussion and engagement.

---

> > ### Comment · Reviewer_AmHM · 2025-11-25
> >
> > Thank you for addressing the questions on the dataset, evaluation metrics, and computational cost. However, the core methodological concerns raised in my initial review remain unresolved (in Weaknesses). Therefore, I will keep my score unchanged.

---

### Official Review · Reviewer_fs7m · 2025-11-04

**Soundness:** 3
**Presentation:** 2
**Contribution:** 3
**Rating:** 6
**Confidence:** 3

**Summary:**

This paper proposes a multi-agent retrieval enhancement framework, MARWA, to address the robustness and scalability issues in bioinformatics workflow automation.
With the rapid growth of multi-omics data, bioinformatics analysis workflows are becoming increasingly complex, and manually constructing workflows is both time-consuming and error-prone.
Existing automation methods based on Large Language Models (LLMs) suffer from issues such as "one-off generation," inaccurate tool retrieval, and insufficient evaluation. MARWA significantly improves the accuracy of tool retrieval and command generation by employing six collaborative agents (analysis, planning, selection, generation and execution, debugging, and judgment), combined with retrieval enhancement (RAG), multi-perspective LLM tool description, and comparative learning.
Furthermore, the paper proposes a two-stage evaluation system that combines expert execution and large-scale LLM evaluation. Experiments demonstrate that MARWA outperforms existing methods across pass rate, workflow quality, and scalability, laying the foundation for reliable bioinformatics automation workflows.

**Strengths:**

- This work employs a multi-agent collaborative architecture, involving a series of intricate processes including analysis, planning, selection, execution, debugging, and judgment, thereby significantly improving process robustness and flexibility.

- The method uses LLM to generate multi-perspective tool descriptions and optimizes tool embedding through BERT contrastive learning, achieving tool retrieval accuracy higher than mainstream baselines.

- Real-world execution and large-scale evaluation are combined: 40 expert validation tasks and 2270 LLM evaluation tasks provide a comprehensive evaluation system, making the results highly convincing.

**Weaknesses:**

- While the evaluation system is comprehensive, large-scale tasks primarily rely on automated evaluation using LLM, resulting in a limited number of actual tasks executed. Furthermore, for biomedicine, does over-reliance on automated evaluation accurately reflect real-world usability?

- The tool's database expansion primarily relies on manual verification and command logging, leaving room for improvement in automation and scaling up.

- The detailed description of contrastive learning is rather brief, lacking sufficient disclosure of hyperparameters, training set size, and other details.

**Questions:**

- Will the multi-perspective generation of tool descriptions in MARWA's search enhancement feature lead to semantic drift due to LLM illusion? How can description consistency be guaranteed?

- Are there compatibility solutions for the file system interface across different operating systems (e.g., Windows, macOS)? How general is its practical deployment?

- In large-scale evaluations, is there more detailed statistical analysis of the accuracy of LLM automatic scoring and its consistency with expert scoring?

- How adaptable is MARWA to new tools or parameter changes? Does it support automatic tool version identification and compatibility?

- Will multi-agent collaboration lead to significant computational resource consumption? What are the actual hardware requirements for deployment?

---

> ### Author Response · Authors · 2025-11-18
>
> Dear Reviewer,
>
> Thank you for these helpful comments. Our detailed answers are provided below.
>
> > Q1: Will the multi-perspective generation of tool descriptions in MARWA's search enhancement feature lead to semantic drift due to LLM illusion? How can description consistency be guaranteed?
>
> Thank you for this important point.
> We prevent LLM "hallucination" and ensure description consistency through two main safeguards: We don't let the LLM generate descriptions from scratch. We provide it with the **original, ground-truth description** for a tool.
> Our experimental results **(Seaction3.2.1, Table 1)** show that our method significantly outperforms other state-of-the-art models. This is direct evidence that our generated descriptions are accurate and consistent, successfully capturing each tool's true function.
>
> > Q2: Are there compatibility solutions for the file system interface across different operating systems (e.g., Windows, macOS)? How general is its practical deployment?
>
> Thank you for your important question regarding cross-platform compatibility and deployment generality.
>
> Our file system interface is **fully compatible** with major operating systems like Windows, macOS, and Linux. This is achieved by using Python's standard libraries, which automatically handle platform-specific details like path separators. This allows our framework to run seamlessly across different systems without any code modification.
>
> > Q3: In large-scale evaluations, is there more detailed statistical analysis of the accuracy of LLM automatic scoring and its consistency with expert scoring?
>
> Thank you for the excellent and crucial question.
> Directly verifying LLM scoring accuracy against experts on thousands of tasks is impractical. Our approach was to rigorously **validate the LLM judges on our expert-scored small dataset first, before applying them at scale**.
>
> To quantify this, we established metrics like the **PFAR** and **SAR**. We found that the LLM's scores are highly consistent with those of human experts. As detailed in **(Section 4.4 and Appendix D.5)**, our analysis shows consistently strong agreement, with **PFAR values often exceeding 0.90** and **SAR values approaching 0.90**. This provides strong statistical evidence of their scoring alignment.
>
> Because the LLMs demonstrated such high, validated accuracy on this representative sample, we were confident in applying them as reliable evaluators for our large-scale benchmark.
>
> > Q4: How adaptable is MARWA to new tools or parameter changes? Does it support automatic tool version identification and compatibility?
>
> Thank you for your question. MARWA's adaptability is a core part of its design, enabling it to handle the dynamic bioinformatics tool ecosystem in three key ways:
>
> **1. Learning New Tools:** When a task requires a tool not in our database, MARWA attempts to use it based on the LLM's general knowledge. If this fails, our Debugging Agent automatically generates and tests installation commands. Once a working command is found, it is automatically added to our knowledge database, allowing MARWA to learn the new knowledge.
>
> **2. Adapting to Parameter Changes:** If a tool update changes a command-line parameter, the old command will fail. The Debugging Agent analyzes the error message, corrects the parameter, and retries. This self-correcting loop ensures workflows remain functional even after tool updates, and the validated command updates our database.
>
> In summary, MARWA uses a robust **"execute, fail, correct, and learn"** cycle to dynamically adapt to new tools, parameter changes, and versioning, ensuring long-term reliability.
>
> > Q5: Will multi-agent collaboration lead to significant computational resource consumption? What are the actual hardware requirements for deployment?
>
> Thank you for raising this critical question.
>
> **1. Regarding Computational Resource Consumption:**
>
> We have added a **Cost-Effectiveness Analysis (Section 4.6, Table 5)** and calculated the **Effective Cost Per Success (ECPS)**, which is the average dollar cost for one successful workflow.
>
> | Method  | ECPS  |
> | --------- | ----- |
> | LLM-only  | 0.825 |
> | AutoBA   | 0.383 |
> | ReAct      | 0.521 |
> | BioMaster | 0.693 |
> | MARWA    | **0.310** |
>
> The results demonstrate that **MARWA** achieves the lowest cost of all methods, with an **ECPS of just $0.310**. While MARWA's multi-agent process may appear more complex than a single-shot approach, it is **significantly more resource-efficient in practice**.
>
> **2. Regarding Hardware Requirements:**
>
> The most computationally intensive task, LLM inference, is handled via API, consuming no high-end local hardware. The local machine only needs to run the lightweight agent orchestration and retrieval modules.
>
> Thank you again for your great efforts and valuable comments. If you have any further questions or require additional clarification, we welcome continued discussion and engagement.

---

### Official Review · Reviewer_SPqT · 2025-11-04

**Soundness:** 2
**Presentation:** 2
**Contribution:** 3
**Rating:** 4
**Confidence:** 4

**Summary:**

This paper presents an innovative Multi-Agent Retrieval-Augmented framework aimed at enhancing bioinformatics workflow automation. The approach leverages multi-perspective LLM-enhanced tool descriptions combined with contrastive representation learning to achieve robust semantic representations of bioinformatics tools, ultimately improving tool retrieval accuracy. The evaluation dataset constructed for this purpose could serve as a valuable resource for the research community.

**Strengths:**

1.	The integration of multi-perspective LLM-enhanced tool descriptions is a promising approach for tool selection in complex scientific domains, potentially benefiting agent systems.
2.	The two proposed datasets could significantly aid in the evaluation of bioinformatics agents.

**Weaknesses:**

1.	The claim of complete automation in the current workflow seems somewhat overstated. Is there any algorithmic illustration provided? Is the sequence of operations predefined?
2.	The framework includes six cooperative LLM-based expert agents. Are the same models used across all six components, or are there distinct characteristics for different experts? Insights into model selection would be beneficial.
3.	There is a sentence structure issue on lines 263-264 that needs clarification.
4.	Is the file system intended to be multimodal?
5.	Which specific LLMs are used as evaluators? What distinguishes the evaluation process from the judging operation?
6.	A more detailed presentation of the dataset's difficulty and characteristics would enhance clarity. Additionally, what are the cost differences between MARWA and test-time scaling with powerful LLM-only methods in solving the task?

**Questions:**

Identical to the 'Weaknesses' noted

---

> ### Author Response · Authors · 2025-11-18
>
> Dear Reviewer,
>
> Thank you for these helpful comments. Our detailed answers are provided below.
>
> > Q1: The claim of complete automation in the current workflow seems somewhat overstated. Is there any algorithmic illustration provided? Is the sequence of operations predefined?
>
> Thank you for this important point. To clarify, **the sequence of operations is not predefined but is dynamically determined by our agents in a closed-loop system.** The next step depends on the analytical goal, the current workflow state, and the file system. To make this explicit, **we have added a detailed algorithm in the Appendix B**.
>
> > Q2: The framework includes six cooperative LLM-based expert agents. Are the same models used across all six components, or are there distinct characteristics for different experts? Insights into model selection would be beneficial.
>
> Thank you for this insightful question. We use the same backbone model (GPT-4 Turbo) for all six agents. We chose this design for **experimental fairness,** allowing us to prove that our performance gains come from our **framework's architecture** rather than the model itself. We have clarified this in **Section 4.4**.
>
> However, your question about model selection is highly pertinent for future development. We believe that **exploring a heterogeneous model setup is a promising direction for future work.**
>
> > Q3: There is a sentence structure issue on lines 263-264 that needs clarification.
>
> We sincerely thank you for catching this grammatical error. We have located the sentence in question and have rewritten it for better clarity.
>
> > Q4: Is the file system intended to be multimodal?
>
> Thank you for this insightful question about the scope of our file system interface. In its current implementation, the file system interface is **not multimodal**. It operates at a **textual metadata level**, providing the agents with information about the *existence*, *names*, *formats*, and *paths* of files and directories within the execution environment.
>
> > Q5: Which specific LLMs are used as evaluators? What distinguishes the evaluation process from the judging operation?
>
> Thank you for this important question.
>
> **1. LLM Evaluators:** We used three specific models as external evaluators: **GPT-4o, Gemini-2.5-pro-exp, and Qwen 2.5 72B-Instruct**. This helps ensure objective scoring. We have listed them in **Section 4.2**.
>
> **2. Judging vs. Evaluation:** These are two distinct functions:
>
> - **The Judging Agent** is **internal** to MARWA. It runs *during* workflow generation and decides if the task is complete, telling the system whether to stop or continue. It’s part of the **automation engine**.
> - **The Evaluation Process** is **external** and used for our experiments. It runs *after* a workflow is generated to score its final quality against our benchmarks. It's part of our **performance measurement**.
>
> > Q6: A more detailed presentation of the dataset's difficulty and characteristics would enhance clarity. Additionally, what are the cost differences between MARWA and test-time scaling with powerful LLM-only methods in solving the task?
>
> Thank you for this excellent suggestion.
>
> **1. Dataset:** Thank you for this excellent point. We agree that characterizing task difficulty is very important. Our main priority with the current dataset was to ensure **broad coverage** across the key fields of bioinformatics (genomics, transcriptomics, etc.) to test our framework's versatility. You've rightly highlighted that the next important step is to analyze the **depth and difficulty** of these tasks.
>
> **2. On Cost Differences vs. Powerful LLM-only Methods:** We added a **Cost-Effectiveness Analysis (Section 4.6, Table 5)** and calculated the **Effective Cost Per Success (ECPS)**, which is the average dollar cost for one successful workflow.
>
> | Method    | I-Tokens (S) | O-Tokens (S) | I-Tokens (F) | O-Tokens (F) | ECPS  |
> | --------- | ------------ | :----------- | ------------ | ------------ | ----- |
> | LLM-only  | 298          | 689          | 1396         | 2836         | 0.825 |
> | AutoBA    | 945          | 1043         | 1373         | 2218         | 0.383 |
> | ReAct     | 4085         | 1329         | 6526         | 1963         | 0.521 |
> | BioMaster | 6972         | 1920         | 9203         | 3221         | 0.693 |
> | MARWA     | 5117         | 1831         | 6529         | 2357         | **0.310** |
>
> The results demonstrate MARWA's significant cost superiority:
>
> - The **LLM-only** method, despite its apparent simplicity, is extremely expensive in practice, with an **ECPS of $0.825**. Its low success rate means one must pay for numerous failed, costly API calls for every one successful outcome.
> - In stark contrast, **MARWA** achieves the lowest cost of all methods, with an **ECPS of just $0.310**.
>
> Thank you again for your great efforts and valuable comments. If you have any further questions or require additional clarification, we welcome continued discussion and engagement.

---

### Meta-Review · Area_Chair_78Jg · 2026-01-04

**Summary:**

The reviewers' concerns primarily focused on the paper's methodological novelty, domain specificity, and evaluation rigor.

Reviewer AmHM and Reviewer CtBg questioned the conceptual innovation, viewing the framework as an incremental "layered adaptation" of existing paradigms (e.g., RAG, step-by-step generation) rather than a fundamental algorithmic breakthrough or a deeply domain-optimized architecture .

Concerns regarding evaluation were also prominent. Reviewer fs7m questioned the reliability of using LLMs for large-scale automated scoring , while Reviewer CtBg and Reviewer AmHM initially flagged the lack of cost-effectiveness analysis compared to general-purpose baselines .

Additionally, Reviewer CtBg raised technical doubts about the choice of fine-tuning a BERT backbone over modern embedding models , and Reviewer SPqT sought clarification on the distinction between internal judging agents and external evaluation protocols.

**Reviewer Concerns:**

While the rebuttal successfully addressed empirical questions raised by Reviewer SPqT, Reviewer fs7m, and Reviewer CtBg by providing a convincing cost-effectiveness analysis (ECPS), validating the efficiency of the fine-tuned BERT model against modern baselines, and demonstrating superior performance over general-purpose agents , critical issues remain outstanding.

 Reviewer AmHM maintained that the core concern regarding novelty was not resolved , and deeper scrutiny suggests potential weaknesses in the rebuttal itself: the comparison between the fine-tuned BERT and zero-shot modern embeddings may be methodologically unfair; the reliance on LLM-based evaluation for large-scale tasks carries statistical risks of "shared hallucinations" despite small-scale validation; and the claimed domain specificity appears to stem more from engineering configurations than from a deep integration of biological reasoning or knowledge graphs.

**Reviewer Scores:**

Reviewer SPqT: The score would likely remain at 4 or only marginally move to a higher score. While the authors provided the requested cost analysis and pseudocode, this reviewer may still consider the paper’s central claim of "reliable full automation" to be overstated. They might view the system as a practical engineering assembly rather than a scientific breakthrough, maintaining that the paper falls short of the rigorous standards for a top-tier conference despite the additional data.

Reviewer fs7m: The score would likely remain at 6 or increase to 7. This reviewer was already leaning towards acceptance. The authors reinforced their positive stance by providing empirical evidence (Table 1) to show that the LLM-augmented descriptions improved retrieval accuracy rather than causing hallucinations, and by verifying the reliability of the automated evaluation metrics.

Reviewer AmHM: The score would likely remain at 4. Despite the authors adding missing baselines and cost data, this reviewer explicitly stated after the rebuttal that their core concern regarding methodological novelty remained unresolved. They viewed the framework as an incremental "layered adaptation" of existing paradigms (like RAG and ReAct) rather than a fundamental innovation.

Reviewer CtBg: The score would likely remain at 4 or strictly move to a higher score. Although the rebuttal defended the technical choices (fine-tuned BERT, ReAct comparison) from an efficiency standpoint, this reviewer explicitly aligned with Reviewer AmHM regarding the lack of conceptual novelty. They would likely interpret the new experiments as proof of engineering utility rather than research contribution, making them hesitant to champion the paper for acceptance without a stronger theoretical framework.

---

### Decision · Program_Chairs · 2026-01-26

Reject